

# Subsea permafrost and associated methane hydrates: how long will they survive in the future?

Valentina V. Malakhova[1,2] and Alexey V. Eliseev[2,3,4]

[1]Institute of Computational Mathematics and Mathematical Geophysics, Siberian Branch of the Russian Academy of Sciences, Novosibirsk, Russia
[2]Physical Faculty, Lomonosov Moscow State University, Moscow, Russia
[3]A.M. Obukhov Institute of Atmospheric Physics, Russian Academy of Sciences, Moscow, Russia
[4]Kazan Federal University, Kazan, Russia

**Correspondence:** V.V Malakhova (malax@sscc.ru)

**Abstract.** We performed simulations with SMILES (the Sediment Model Invented for Long-tErm Simulations) for 100 kyr in future forced by the output of an Earth System model with internally calculated ice sheets. This Earth System model was driven by idealised scenarios of $CO_2$ emissions (applied at time instant $t = 0$ loosely corresponding to common era year 1950) and by changes of the parameters of the Earth orbit. The simulations are carried out with different values of the heat flux from

the Earth interior. We neglected possible impact of hydrostatic pressure changes due to future sea level changes on freeze/thaw temperature and on thermodynamic stability of methane hydrates. We found that at the outer shelf permafrost disappears either before $t = 0$ or during few centuries in future. In contrast, for the middle and shallow parts of the shelf, in the $CO_2$-emission forced runs the subsea permafrost survives, at least, for 5 kyr after the emission onset or even for much longer. Without an applied greenhouse forcing permafrost exists here at least until 22 kyr after the $CO_2$ emission onset or even survives till the

end of the model runs. At the same parts of the shelf, methane hydrate stability zone disappears not earlier that at $t = 3\,\mathrm{kyr}$ after the $CO_2$ emission onset, but, typically, it survives until 11 to 41 kyr after this onset. Time instants of local extinction of both the subsea permafrost and methane hydrates stability zone (MHSZ) are negatively correlated with the geothermal heat flux because of both permafrost thaw and MHSZ shrinking basically occurs from bottom. However, thaw from the top and the deepening of the MHSZ table is basically determined by the applied $CO_2$ forcing scenario; this is more important for the

permafrost than for MHSZ. In general, the $CO_2$-induced warming in our simulations is able to enhance the pan-Arctic subsea permafrost loss severalfold during 1 kyr after the emissions onset, but is less instrumental for the respective MHSZ loss.

## 1   Introduction

The methane hydrates at the contemporary Arctic shelf are believed to develop during the glaciations of the Pleistocene, when sea level was substantially lower than the present-day one, and this shelf was in a direct contact with a cold atmosphere

(MacDonald, 1990; Buffett, 2000; Romanovskii et al., 2005; O'Connor et al., 2010; Shakhova et al., 2019). This contact allowed for aggrading the permafrost in the exposed shelf (Romanovskii et al., 2005; Portnov et al., 2014; Majorowicz et al., 2012; Angelopoulos et al., 2020), thus providing necessary conditions for thermodynamic stability of methane hydrates within



the so called methane hydrate stability zone (MHSZ). Both the subsea permafrost and the permafrost-associated methane hydrates (PAMH) survived until the present owing to their long, of the order of $10^1$ kyr (Romanovskii et al., 2005; Malakhova and Eliseev, 2017, 2020a), response time scales to temperature anomaly at the top of the sediments.

A natural question arises is how long the subsea permafrost and PAMH would survive given the ongoing climate warming and future changes in the parameters of the Earth orbit. While temperature changes at the sea floor at the shelf is projected to be rather modest at the century timescale even under relatively strong anthropogenic warming (Lamarque, 2008), a stronger warming may occur at a larger time scale. One possible mechanism for this is due to the riverine export of sensible heat (Dmitrenko et al., 2011; Golubeva et al., 2018; Shakhova et al., 2019). Even more obvious albeit likely less efficient mechanism is a direct heating of the oceanic water from above. As a result, at millennium and longer timescales, the thermal state of the subsea sediments may be changed markedly with respective impacts on the subsea permafrost and PAHM MHSZ. For instance, Wilkenskjeld et al. (2021) projected an accelerated degradation of the subsea permafrost during the next millennium. Further, in simulations by Hunter et al. (2013) until 2850 C.E. (common era), methane release from the hydrate dissociation either accelerates during the incoming millennium or exhibits a peak followed by a decline depending on the applied warming scenario. Even longer, 20 kyr and 100 kyr-scale, future projections were performed, correspondingly by Majorowicz et al. (2012) and by Archer (2015).

All these projections except that by Majorowicz et al. (2012) neglect future changes in parameters of the Earth orbit which evolution might initiate new glacial inception. It was simulated earlier that both the subsea permafrost and PAHM MHSZ were markedly thinner or even non-existent at the middle and outer parts of the Arctic shelf aftermath the Pleistocene glacial terminations (Romanovskii et al., 2005, 2006; Majorowicz et al., 2012; Malakhova and Eliseev, 2017, 2018, 2020a, b; Gavrilov et al., 2020). This inception, in turn, may be delayed because of the ongoing, mostly $CO_2$-induced climate warming (Archer and Ganopolski, 2005; Ganopolski et al., 2016). Thus, it is of vital interest to study the impact of these two forcings combined.

A related problem is due to the measurable present day dissociation of hydrates with the gas venting out from the sediments to the water and further from the water to the atmosphere (Buffett, 2000; Romanovskii et al., 2005; O'Connor et al., 2010; Shakhova et al., 2010; Anisimov et al., 2012; Majorowicz et al., 2015; Chuvilin et al., 2018; Shakhova et al., 2019). Methane hydrates dissociation was also inferred from the isotopology measurements as a possible cause for development of past hot epochs (e.g., the Paleocene-Eocene Thermal Maximum, Dickens et al., 1995; Zeebe, 2012). Large spatial and seasonal variability associated with these emissions hamper even the present-day pan-Arctic estimates of this source. In particular, while most studies, including the Intergovernmental Panel On Climate Change Working Group 1 Sixth Assessment Report (IPCC WG1 AR6), conclude that total methane flux from the surface of all (including both the Arctic Ocean and other oceans) shelf areas to the atmosphere is $\leq 10\,\mathrm{Tg\,CH_4\,yr^{-1}}$ (Saunois et al., 2020; Canadell et al., 2021), there are claims that this flux may be markedly larger (e.g., Shakhova et al., 2010).

These fluxes might become much stronger near the timing of complete local extinction of the permafrost and hydrate layers, especially given the upper estimate of the methane stock in the submerged permafrost-associated hydrates of $1400\,\mathrm{PgC}$ (Shakhova et al., 2010; James et al., 2016). A necessary ingredient for such enhancement is an accumulation below the frozen sediment layer of the methane from the hydrates dissolved in the lower part of MHSZ (Majorowicz et al., 2012; Sapart et al.,





2017). This trapped gas waits until the impermeable layer disappears with a pulse release of methane aftermath. Despite the latter phenomenon may be suppressed by tentative existence of channels because of preformed taliks in the submerged pale-
orivers valleys and lagoons, especially in regions with high geothermal heat flux (Frederick and Buffett, 2014; Majorowicz et al., 2015; Malakhova and Eliseev, 2018; Angelopoulos et al., 2021), any information on the timing of such potentially catastrophic release is of prominent interest.

  This goal is pursued in the present paper. We employ idealised simulations with a model for sediment thermophysics forced by the most relevant climate forcings: slow variations due to evolution of the Earth orbit and due to a century-scale anthro-
pogenic greenhouse warming followed by a relaxation of the atmospheric $CO_2$ content to a new equilibrium value. Based on these simulations, we estimate how long the submerged subsea permafrost and PAHM MHSZ will survive in future. Our simulation setup lacks an explicit geography, but observes the dependence of the climate forcing in the Pleistocene on the contemporary shelf depth. The latter dependence is reflected in lengths of time intervals when shelf is either inundated or exposed to the atmosphere and is an essential ingredient for resolving the typical behaviour of different parts of the Arctic shelf
responding to the climatic forcing at century, millennium and longer timescales.

## 2   Model and simulations

We use the version of the SMILES (the Sediment Model Invented for Long-tErm Simulations) which is identical to that described in (Malakhova and Eliseev, 2020b). It has evolved from the model for sediment thermophysics (Denisov et al., 2011; Eliseev et al., 2015; Malakhova and Golubeva, 2016; Malakhova and Eliseev, 2017, 2018, 2020a) by extending the
earlier model version with an equation for salt diffusion in the sediment pores. In brief, the model solves two coupled one-dimensional diffusion equations: one is for heat diffusion in the sediments (taking into account heat which is consumed during thaw or released during freezing) and another one is for salt diffusion in the sediments. Both equations are solved in the sediment column of depth $H_S = 1,500\,\mathrm{m}$. For heat diffusion equation, a condition of temperature continuity are imposed at the thaw/freezing interfaces in the sediments; this condition is coupled to the Stefan condition. For the same equation, heat
capacity and thermal conductivity depend on the state of the sediment layer (either frozen or unfrozen; Table S1). Salt diffusion is allowed in the unfrozen layers only. Freeze/thaw temperature depends on salinity and pressure allowing for coupling between heat and salt diffusion equations. The latent heat of fusion during formation and melting of the pore ice is explicitly accounted for, but the respective heat released during dissociation of hydrates is neglected. Sediment porosity $\phi$ exponentially decreases downward from the value 0.4 at the top of the sediments with the vertical scale 2,500 m.
Equilibrium methane hydrates stability boundary is adapted from the TOUGH+HYDRATE model taking into account salt-induced depression of the dissociation temperature (Reagan and Moridis, 2008; Reagan et al., 2011).

  More detailed description of SMILES is available in the supplement (Sect. S1).

  Initial conditions for both heat and salt diffusion equations are applied for a non-glacial state as it is reconstructed for 400 kyr B.P. (before present). Here 'present' (or time instant $t = 0$) is formally ascribed to year 1950 C.E. The employed initial
conditions are described in more details in the supplement (Sect. S2).



At the sediment–ocean interface (or at the sediment–air interface if the sediments are in contact with the air during oceanic regressions), temperature and salinity are prescribed to time-dependent functions $T_B$ and $S_B$.

For past time instants and for the present day ($t \leq 0$) , when shelf is in contact with the atmosphere, $T_B$ is set equal to surface air temperature (SAT) $T_a$, and $S_B$ is zeroed. When shelf is covered by water, $T_B$ ($S_B$) is prescribed to be equal to the

near–bottom water temperature (salinity) $T_w$ ($S_w$). Both $T_w$ and $S_w$ are functions of the present-day shelf depth $H_D$ (Table S3). At the bottom of sediment domain, time-independent heat flux $G$ from the Earth interior and no-flux condition for salinity are adapted. The time-dependent $T_a$ is constructed from the monthly mean SAT simulated with the Climber-2 for time interval from 400 kyr B.P. to $t = 0$ (Ganopolski et al., 2016) as it is detailed in supplementary Sect. S3.

Then, our simulations are continued for another 100 kyr. We loosely refer this time interval as a 'future' ($t > 0$) and mark

it with 'after present' (A.P.). In this, we assume that the shelf is always covered by water but SAT changes. Thus, for future $T_B = T_w + \Delta T_{fut}$. In the first series of simulations, $T_{fut}$ is set equal to $T_a(t) - T_a(0)$. For this, we use the continuation of the Climber-2 simulations forced by changes of parameters of the Earth orbit and by anthropogenic $CO_2$ emissions (Ganopolski et al., 2016). These emissions start in year 1950 C.E. and proceed with a simulation-independent rate until the prechosen cumulative emission level $E_{tot}$ is achieved. We chose two Climber-2 simulations, one with $E_{tot} = 1000\,PgC$ and another

with $E_{tot} = 3000\,PgC$, in which emissions cease in future years 100 and 300 correspondingly. Upon this, anthropogenic $CO_2$ emission rate is set to zero, and the Climber-2 simulation is continued with a freely evolving carbon cycle. Thereafter, our simulations based on the Climber-2 output with $E_{tot} = 1000\,PgC$ and $E_{tot} = 3000\,PgC$ are referred to as TR1000 and TR3000, respectively.

However, the sea floor warming would likely proceed in a much slower rate than the surface air warming (Lamarque,

2008) except over the shallowest part of the shelf (up to few metres, Dmitrenko et al., 2011; Overduin et al., 2019). Thus, the change of the surface air temperature is a poor proxy for temperature change at the sea floor. Therefore, we performed one more, 'committed' simulation, in which future $T_a(t)$ is a repetition of the Climber-2-simulated $T_a(0)$ (corresponding to $\Delta T_{fut} \equiv 0$). This simulation is thereafter referred to as TR0. In brief, our simulations represent a 'window of possibilities' with the temperature change in TR3000 (TR0) serving as an uppermost (lowermost) possible sea floor warming.

We highlight that anthropogenic emissions in TR1000 and TR3000 attempt to mimic neither historical emissions nor common scenarios for anthropogenic emissions in future (e.g., Eyring et al., 2016). Nonetheless, taking into account that cumulative anthropogenic $CO_2$ emissions into the atmosphere for 1750-2004 are close to 500 PgC (Friedlingstein et al., 2020), one could loosly ascribe model year 50, when the cumulative emissions are close to this value, to year 2000 C.E. This ascription is not principal to our results, but might be helpful for putting the figured numbers into the context of the ongoing and future climate

changes.

For future period, we neglect sea level changes on hydrostatic pressure. The impact of such assumption is discussed in Sect. 4. A neglect of pressure contribution is also characteristic for some of the other estimates of future methane hydrate response to climate changes (Buffett and Archer, 2004; Hunter et al., 2013).

The value of the heat flux from the Earth interior $G$ is time-independent, but is varied between different simulations. De-

pending on simulation, we set it equal to either $45\,mW\,m^{-2}$ or to $60\,mW\,m^{-2}$ or to $75\,mW\,m^{-2}$. The intermediate of these





values is characteristic for the most part of the Arctic shelf (Pollack et al., 1993; Davies, 2013). In turn, value $G = 75 \, \mathrm{mW \, m^{-2}}$ is typical for rift zones. Regions with $45 \, \mathrm{mW \, m^{-2}}$ are rare in the Arctic, but this value is still studied for completeness.

The setup of our simulations is somewhat similar to that employed by Archer (2015). The difference are i) his model is more detailed than ours (in particular, via explicit treatment of the sediment geochemistry), ii) different $CO_2$ scenarios are applied
(for instance, the cumulative $CO_2$ release into the atmosphere in his driving dataset is $5,000 \, \mathrm{PgC}$), and iii) Archer (2015) accounts for future sea level rise. On the other hand, he disregards future evolution of the Earth orbit, while the orbital forcing is explicitly taken into account in the Climber-2 simulations which are used in our paper.

Another similar simulation is by Majorowicz et al. (2012), who, in addition to that published in (Archer, 2015), implicitly accounted for impact of future changes of the Earth orbit by simply assuming that new glacial inception will occur in 12 kyr.
This approach apparently prolongs future existence of the subsea permafrost and PAHM relative to our results (see below).

Methane content per unit volume of the sediments of the subsea hydrates is calculated via (Gornitz and Fung, 1994; Biastoch et al., 2011; Majumdar and Cook, 2018; Stranne et al., 2017)

$$\tilde{m}_{CH_4} = k_{CH_4} \rho_{CH_4} \phi \theta_{CH_4} \tag{1}$$

where the gas expansion coefficient from the sediment condition to the standard temperature and pressure (STP) is $k_{CH_4} = 140$,
methane density at STP is $\rho_{CH_4} = 0.7168 \, \mathrm{kg \, m^{-3}}$ and $\theta_{CH_4} = 0.05$ is fraction of pore volume occupied by hydrates (Gornitz and Fung, 1994; Buffett and Archer, 2004; Klauda and Sandler, 2005). Then, total methane content per unit area of the sediments, $m_{CH_4}$, is calculated by integrating $\tilde{m}_{CH_4}$ over the estimated methane hydrate stability zone.

## 3 Results

### 3.1 Permafrost

Similar to that obtained earlier with SMILES (Malakhova and Eliseev, 2017, 2018, 2020a, b), a thick permafrost develops in the Arctic shelf to $t = 0$ (Fig. 1). The only exception is case $\left( H_D = 100 \, \mathrm{m}; G = 75 \, \mathrm{mW \, m^{-2}} \right)$, in which permafrost disappears several kiloyears before this time instant. As it is expected, the permafrost layer thickness is larger for simulations with smaller contemporary shelf depth, which is at a longer contact with the atmosphere during oceanic regressions, and for simulations with a smaller geothermal heat flux. Among the studied cases, the largest present-day permafrost layer thickness, about $1,200 \, \mathrm{m}$, is
simulated for case $\left( H_D = 10 \, \mathrm{m}; G = 45 \, \mathrm{mW \, m^{-2}} \right)$.



**Figure 1.** Frozen sediment layers in different SMILES simulations (colours) as functions of time in future for different present-day shelf depths $H_D$ and geothermal flux $G$. Ordinates show the depth below the sea floor. Case ($H_D = 100$ m; $G = 75$ mW m$^{-2}$) is not shown because permafrost in this simulation disappears before $t = 0$.





Upon the start of the $CO_2$–induced warming, the subsea permafrost starts to melt, both from the top and from the bottom. The bottom thaw is basically independent from the applied warming scenario except at the outer shelf but depends on both $G$ and $H_D$. If one averages the bottom thaw rate, $v_{pf,b}$, for 1 kyr after the emissions onset, the maximum value, $\approx 16.7\,\mathrm{m\,kyr^{-1}}$, is simulated for the shallow shelf. For this part of the shelf, the bottom thaw rate depends only weakly on geothermal heat

flux intensity. For the middle shelf, the dependence of $v_{pf,b}$ on $G$ is more marked: this rate changes from $\approx 12\,\mathrm{m\,kyr^{-1}}$ for $G = 75\,\mathrm{mW\,m^{-2}}$ to $15\,\mathrm{m\,kyr^{-1}}$ for $G = 45\,\mathrm{mW\,m^{-2}}$. During the most part of the permafrost degradation, the bottom thaw of the frozen layers is a continuation of the earlier melting, which certainly is not related to $CO_2$ emissions, and is a long-term response to the onset of the Holocene (Malakhova and Eliseev, 2017, 2020a). Further in future, $v_{pf,b}$ slows down.

The permafrost table thaw rate $v_{pf,t}$, in contrast, most strongly depends on applied external $CO_2$ emissions into the at-

mosphere. As it is expected, the larger is the emission rate, the faster is the thaw. For TR0, this rate is always smaller than $\approx 1.5\,\mathrm{m\,kyr^{-1}}$ irrespective of $G$ and $H_D$. Such values are quite similar to those exhibited in the past Holocene. A severalfold larger $v_{pf,t}$, typically from about $2\,\mathrm{m\,kyr^{-1}}$ to approximately $13\,\mathrm{m\,kyr^{-1}}$, is simulated in TR1000 and TR3000 for $0 \le t \le 10\,\mathrm{kyr}$ and, by and large, is independent of $G$. In turn, $v_{pf,t}$ depends on $H_D$ and on $CO_2$ emission rate, but in a complicated way because of the dependence of this rate on the state simulated for $t = 0$. In contrast to $v_{pf,b}$, $v_{pf,t}$ changes in time

non-monotonically but preserving the above-mentioned dependencies on $G$ and $H_D$.

Time before permafrost is extinguished at the Arctic shelf strongly depends on all parameters: contemporary shelf depth, geothermal heat flux, and $CO_2$ emission rate (Fig. 2).

For the outer shelf ($H_D = 100\,\mathrm{m}$) permafrost is either disappears before $t = 0$ or is simulated to disappear during few centuries in future provided that $G$ is sufficiently large ($\ge 60\,\mathrm{mW\,m^{-2}}$ in our experiments). Only for the smallest employed

value of the geothermal heat flux permafrost continues to exist in the future with the date of the complete degradation, $t_{pf,end}$, which is 1 kyr A.P. for TR3000, 2 kyr A.P. for TR1000 , and amounts 11 kyr A.P. for TR0. Similar timescales of the permafrost extinction at the East Siberian Arctic Shelf, from 10 to 50 kyr, are obtained by Archer (2015).

In the $CO_2$-emission forced runs and for the middle and shallow parts of the shelf, $t_{pf,end}$ in our simulations is never smaller than 5 kyr A.P. In this part of the shelf and in the simulation with $G = 45\,\mathrm{mW\,m^{-2}}$, $t_{pf,end}$ is as large as 32 kyr A.P. for

TR3000 and even 51 kyr for TR1000. These values are again in agreement with those reported by Archer (2015).

The longest survival of the subsea permafrost is simulated in runs TR0 in the middle and shallow parts of the shelf. In these experiments, $t_{pf,end}$ is $\ge 22\,\mathrm{kyr}$ A.P. or even survives till the end of the simulation.

All our simulations show clear dependence of $t_{pf,end}$ on geothermal heat flux: the larger the flux is, the sooner the subsea permafrost ceases to exist. This is an obvious consequence of the respective dependence of $v_{pf,b}$. The dependence of $v_{pf,t}$ on

future $CO_2$ emission rate leads to the negative correlation of $t_{pf,end}$ with the applied cumulative emissions. In addition, the time of the subsea permafrost disappearance is smaller for larger $H_D$ because the smaller contemporary shelf depth leads to thinner permafrost layer at $t = 0$.





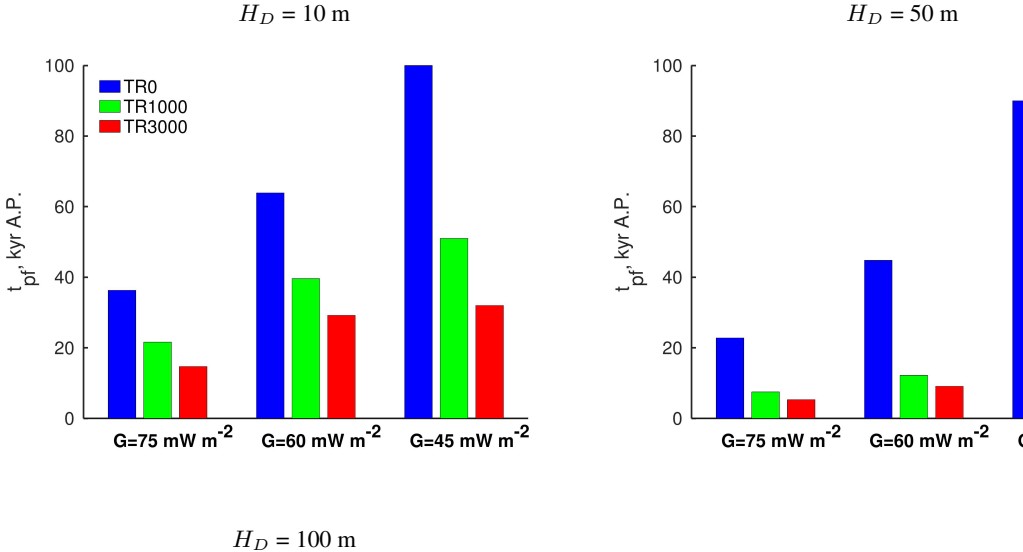

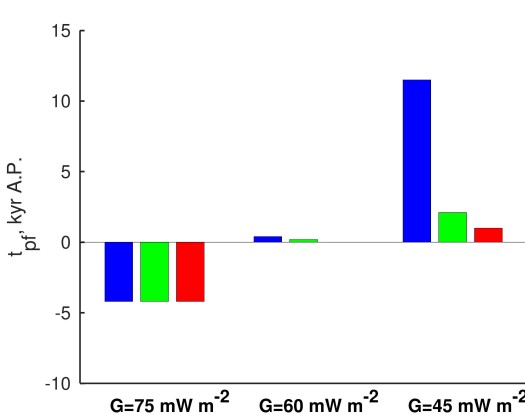

**Figure 2.** Time of the permafrost disappearance. Value 100 kyr A.P. indicates that permafrost survives till the end of the simulation.





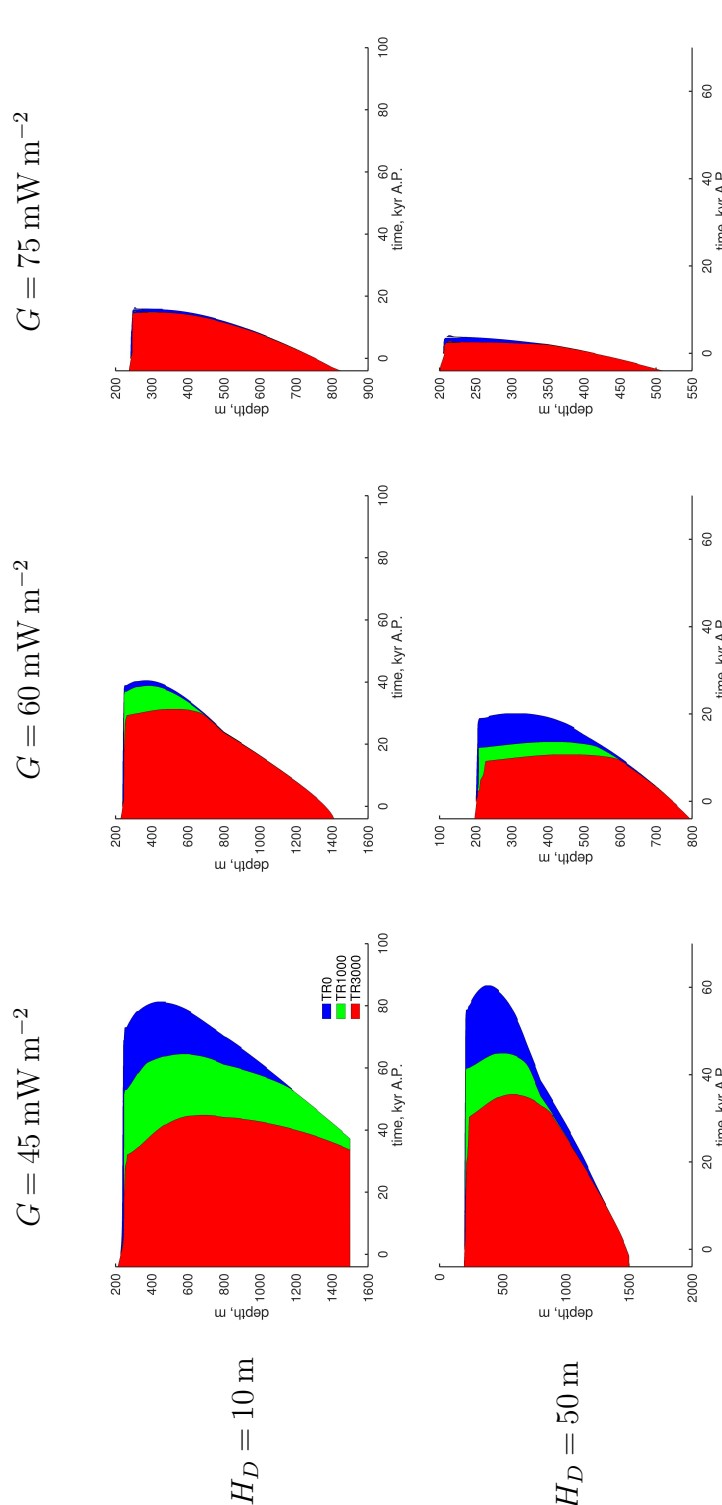

**Figure 3.** Methane hydrate stability zone in different SMILES simulations (colours) as functions of time in future for different present-day shelf depths $H_D$ and geothermal flux $G$. Ordinates show depth below the sea floor. Cases with $H_D = 100$ m are not shown because MHSZ in these simulations disappears before $t = 0$.





## 3.2 Methane hydrates stability zone

The methane hydrate stability zone ceases to exist before the present at the outer shelf ($H_D = 100$ m; Fig. 3). This is in
agreement with our previous simulations (Malakhova and Eliseev, 2017, 2018, 2020a, b) which were driven by other forcing
datasets. For the middle and shallow parts of the shelf in all our simulations, the present-day MHSZ base is located deeper
in the sediments than its permafrost counterpart. This is due to the impact of the hydrostatic pressure of the water. We note,
however, that this impact is only possible in presence of the overlying permafrost layer – otherwise MHSZ does not formed
at all. In turn, the MHSZ top depth is smaller than the permafrost top depth for the same pair $(H_D; G)$. For the middle and
shallow parts of the shelf, MHSZ bottom is located at the about $1,400$ m below the sea floor for $G = 60 \ \mathrm{mW \ m^{-2}}$ and about
$900$ m for $G = 75 \ \mathrm{mW \ m^{-2}}$. In the outer shelf, methane hydrate stability zone disappears before $t = 0$ (Fig. 4c).

After $CO_2$ emission onset, MHSZ starts to shrink. The rate of the shrinking from the bottom, $v_{\mathrm{MHSZ,b}}$, averaged from $t = 0$
to $t = 1$ kyr A.P. amounts from $13 \ \mathrm{m \ kyr^{-1}}$ to $30 \ \mathrm{m \ kyr^{-1}}$ depending on $H_D$ and on $G$, which is similar to its permafrost coun-
terpart. Again, MHSZ shrinking from the bottom is a continuation (albeit slightly fastened) of the corresponding shrinking dur-
ing the last few millennia before the onset of the external $CO_2$ emissions. Later on, $v_{\mathrm{MHSZ,b}}$ magnitude increases. For instance,
its vertical movement rate may be as large as about $100 \ \mathrm{m \ kyr^{-1}}$ for a number of simulations with $H_D = 50$ m. Interestingly,
in simulations with the moderate geothermal heat flux $G = 60 \ \mathrm{mW \ m^{-2}}$ such large values of $v_{\mathrm{MHSZ,b}}$ are exhibited only in the
TR3000 simulation, while in the simulation with the shallowing rate of $v_{\mathrm{MHSZ,b}}$ averaged over $5$ kyr A.P. $\leq t \leq 10$ kyr A.P.
is close to $100 \ \mathrm{m \ kyr^{-1}}$ for all three emission scenarios.

The methane hydrate stability zone, similar to that exhibited for permafrost, shrinks from the top at a much slower rate. On
the shallow and middle parts of the shelf and during the first 1 kyr after the emission onset, this rate, $v_{\mathrm{MHSZ,t}}$, changes from 0
to $6 \ \mathrm{m \ kyr^{-1}}$ depending on contemporary shelf depth, geothermal heat flux, and emission scenario. The MHSZ top deepening
is not a continuation of the tendency during the last few millennia before $t = 0$ – in fact, before the emission onset MHSZ
top shallows rather than deepens at these parts of the shelf in all our simulations. Later on, $v_{\mathrm{MHSZ,t}}$ increases. For the last
few millennia before the complete disappearance of the methane hydrate stability zone, it may be of the order of $100 \ \mathrm{m \ kyr^{-1}}$
(Fig. 3).

The time since the emission onset till the complete MHSZ disappearance (time instant $t_{\mathrm{MHSZ,end}}$) at the shallow and middle
parts of the shelf in our simulations is from 3 kyr for $\left(H_D = 50 \ \mathrm{m}; G = 75 \ \mathrm{mW \ m^{-2}}\right)$ in both runs TR1000 and TR3000
(Fig. 4). For the same pair $(H_D; G)$, it is of the same order of magnitude (4 kyr) in run TR0. However, in other simulations
$t_{\mathrm{MHSZ,end}}$ is located markedly further in future. For $G = 60 \ \mathrm{mW \ m^{-2}}$ and $H_D = 50 \ \mathrm{m} \ (H_D = 10 \ \mathrm{m})$, $t_{\mathrm{MHSZ,end}}$ changes
from 11 to 20 kyr A.P. (from 31 to 41 kyr A.P.) depending on the applied $CO_2$ scenario. For small geothermal heat flux
$G = 45 \ \mathrm{mW \ m^{-2}}$, this time instant is from 36 to 81 kyr. A.P. depending on $CO_2$ scenario and on contemporary shelf depth.
Basically, $t_{\mathrm{MHSZ,end}}$ is negatively correlated with both $G$, $H_D$ and with the rate of the $CO_2$-induced warming in the atmosphere.

## 3.3 Methane release from the sediments to the water

We estimated the release of methane from sediments to the oceanic water based on $dm_{\mathrm{CH_4}}/dt$ (Eq. 1). In this, we assume that



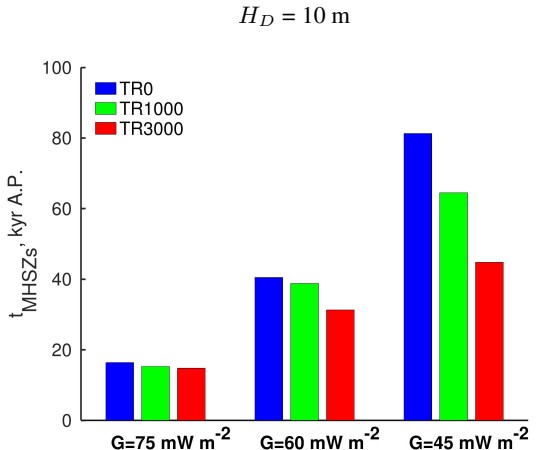

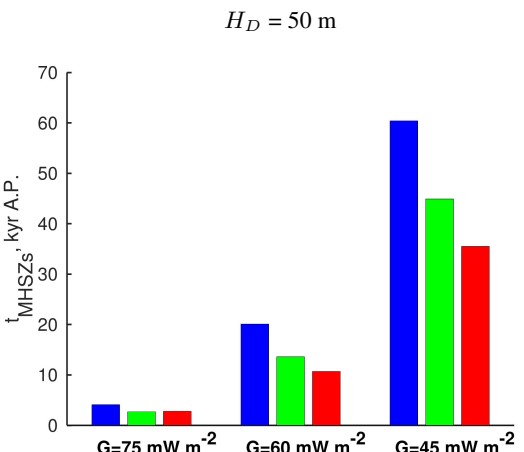

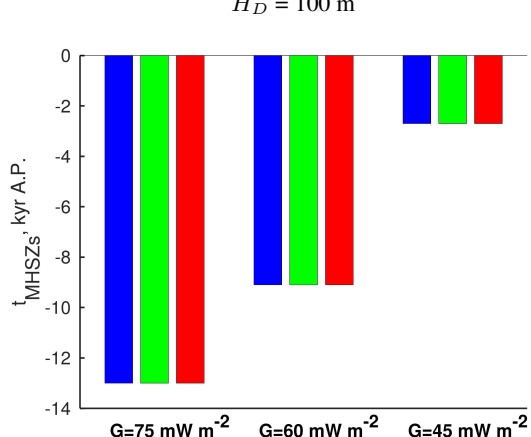

**Figure 4.** Time of the methane hydrate stability zone disappearance.





- Methane, which is released from the hydrate dissociation because of MHSZ shrinking, is instantly transported to the sediment-water interface but is subjected to chemical loss in the sulphate reduction zone. The latter loss is represented via coefficient $K_S < 1$.

- When MHSZ shrinks, the only escaping methane is from the top of the methane hydrate stability zone. Methane produced due to hydrates dissociation at the MHSZ bottom, is accumulated below this zone. However, when MHSZ disappears completely, this accumulated methane is instantly transported to the sediment-water interface.

Thus, the ocean-to-atmosphere $CH_4$ flux per unit area reads

$$f_{CH_4} = K_S \frac{dm_{CH_4}}{dt}. \tag{2}$$

Coefficient $K_S$ is set equal to the spatially uniform value of 0.5, which is adapted from the synthesis (Ruppel and Kessler, 225   2017).

The results averaged over the prechosen time intervals are shown in Fig. 5. For 'recent past' (the last millennium before the $CO_2$ emission onset) for both shallow and middle parts of the shelf is between 1.4 and $3.2\,\mathrm{gCH_4\,m^{-2}\,yr^{-1}}$. In future, $f_{CH_4}$ basically decreases in simulation TR0, which is consistent with the idea that in this experiment the response of the subsea permafrost and the subsea methane hydrates are an adjustment to the onset of the Holocene. In contrast, future en-230   hancement of the MHSZ shrinking from the top in simulations TR1000 and TR3000 leads to overall increase of this flux. In particular, means of $f_{CH_4}$ over 0.5-1 kyr A.P. in these model runs typically amounts from 3 to $7.5\,\mathrm{gCH_4\,m^{-2}\,yr^{-1}}$ at the shallow and intermediate shelves (except for $\left(H_D = 50\,\mathrm{m}; G = 75\,\mathrm{mW\,m^{-2}}\right)$ in experiments TR1000 and TR3000). The most marked, by one or two orders of magnitude, increase of the methane release from the sediments to the water occurs when MHSZ ceases to exit. The largest obtained value is a mean over 2-5 kyr A.P. for which is equal is $125\,\mathrm{gCH_4\,m^{-2}\,yr^{-1}}$ for 235   $\left(H_D = 50\,\mathrm{m}; G = 75\,\mathrm{mW\,m^{-2}}\right)$. This value is insensitive to the applied emission scenario. The latter is a consequence of the dominance of the bottom shrinking in the reduction of the MHSZ thickness for this particular case. An order-of-magnitude smaller peak emission as averaged over 2-5 kyr is simulated for $\left(H_D = 50\,\mathrm{m}; G = 60\,\mathrm{mW\,m^{-2}}\right)$, but only in experiment TR3000. We note, however, this order-of-magnitude difference is an artefact of averaging over longer time interval (either 0.5 kyr or 3 kyr). Peak $CH_4$ release per time step is of the same order of magnitude between $\left(H_D = 50\,\mathrm{m}; G = 75\,\mathrm{mW\,m^{-2}}\right)$ 240   and $\left(H_D = 50\,\mathrm{m}; G = 60\,\mathrm{mW\,m^{-2}}\right)$ (not shown). In addition, very similar peak release of methane is simulated with the same pairs $\left(H_D; G\right)$ irrespective of applied emissions, but at different time instants corresponding to those shown in Fig. 4 – this is again a manifestation of the dominance of the bottom MHSZ shrinking over the top one.

We note that our $f_{CH_4}$ estimates for time interval from -0.5 kyr A.P. to +0.5 kyr.A.P. are within the corresponding range attributed to the subsea permafrost thaw as reported by Shakhova et al. (2019) ($\leq 1\,\mathrm{gCH_4\,m^{-2}\,yr^{-1}}$ in our units). Other 245   methane sources, which, according to Shakhova et al. (2019), which may lead to much larger fluxes, are ignored in our paper.

## 3.4   Implications for the pan-Arctic

Despite of very rudimental account of geographically distributed properties, it is instructive to estimate the pan-Arctic values of the above studied variables. Thus, we integrated our simulated variables over the whole Arctic shelf with a crude approximation





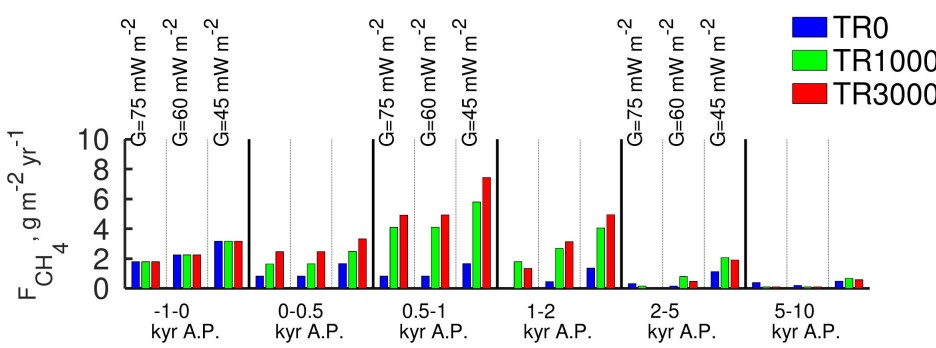

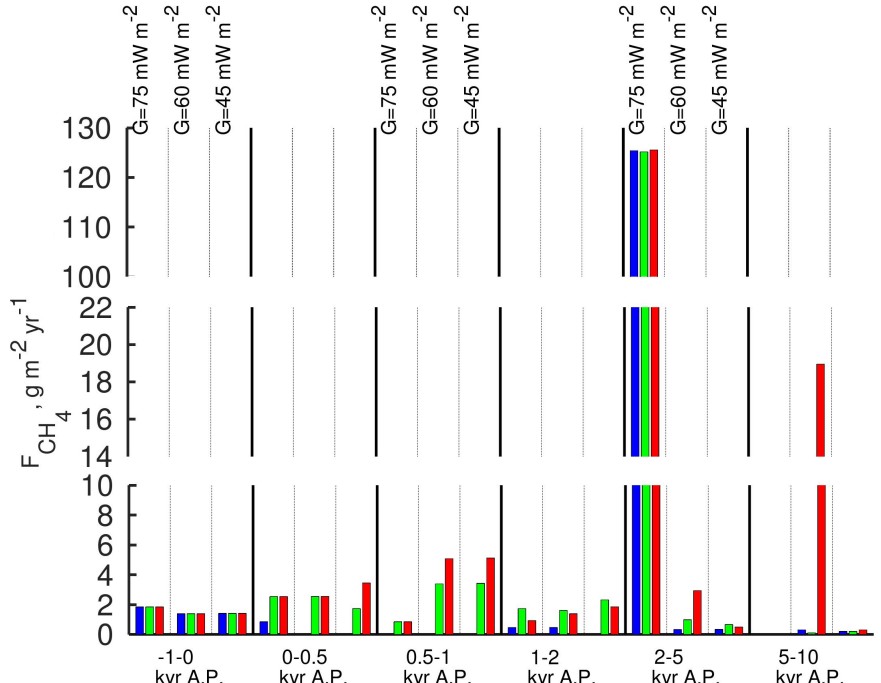

**Figure 5.** Methane flux from the ocean to the atmosphere averaged over different time intervals.





for such integrals: we assume that our shallow shelf is representative for all shelf areas with $H_D \leq 30$ m, our intermediate shelf
– for $30\,\text{m} < H_D \leq 75\,\text{m}$, and our deep shelf – for $H_D > 75\,\text{m}$. To partly compensate for our rudimental geography, we
assume limit these integrals to the regions where the subsea permafrost was simulated in a more realistic setup (Malakhova,
2020) (supplementary Sect. S4), in which the geographically explicit surface forcing was used (supplementary Fig. S3 and
Table S4) – just by multiplying vertical integrals over the area of such regions. We note that the subsea permafrost distribution
in the Arctic simulated by Malakhova (2020) is in general agreement with an alternative simulation (Overduin et al., 2019). We
also assume that MHSZ develops only in the subsea permafrost and covers the whole permafrost-bearing region as simulated
by Malakhova (2020). In addition, we acknowledge following important caveats in our 'pan-Arctic' calculations:

– Geological features are neglected completely. Such features may lead either to local variations of the geothermal heat
flux or to release of the thermogenic methane from the sediments.

– In our simulations, we use the Climber-2 surface air temperature anomaly from the present day only for the grid cell
corresponding to the East Siberian Arctic shelf to for our model. This anomaly is apparently different even from temperature in other model grid cells. However, three shelf regions are located in nearby grid cells (recall very coarse zonal
resolution of Climber-2, $\approx 51^o$), and temperature anomaly in our preselected Climber grid cell deviates from its zonal
mean counterpart no more than by 20% during the most part of the simulation (supplementary Sect. S3 and Fig. S2).

– Moreover, we use geographically uniform value for reference temperature $-12^oC$ to which the Climber-simulated
anomalies are added (supplementary Sect. S3). This caveat is partly (albeit far from completely) ameliorated by using the above-mentioned subsea permafrost extent adapted from (Malakhova, 2020).

We these caveats in mind, we highlight that our calculations are correct only for an order of magnitude.

In addition, because of the aforementioned limitations, we do not make attempt to calculate the present day volumes of the
subsea permafrost and of the permafrost-associated methane hydrate stability zone – both of them may be sensitive to the local
geological features – and present only changes relative to the present day.

The subsea permafrost volume, $V_{pf}$, is decreased by 1.3% (4.5%, 6.8%) during the first 0.5 kyr in future in simulation TR0
(TR1000, TR3000) with $G = 60\,\text{mW m}^{-2}$ (Fig. 6a). Larger permafrost volume loss during the same time interval is simulated
with $G = 75\,\text{mW m}^{-2}$: 2% (10%, 13%). At $t = 1$ kyr A.P., $V_{pf}$ in simulations TR0 is decreased by 1.5-4.6% relative to $t = 0$
depending on $G$, by 4-14% in simulations TR1000, and by 8-20% in simulations TR3000. Our estimates are markedly smaller
those reported by Wilkenskjeld et al. (2021) who claimed that 35% of the initial subsea permafrost volume is lost by common
era year 3000 under high emission scenario SSP5-8.5. While our and theirs modelling setups are pretty different because of
i) differences in emission scenarios, ii) their usage of the Max-Planck-Institute (MPI) oceanic model to generate the temperature
at top of the sediments rather than our simple transfer of the near-surface atmospheric temperature anomaly to the the sediment-oceanic interface, iii) an explicit treatment of the geographic features in their simulations and a very rudimental account for
them in ours, and iv) an incomplete correspondence between our calender and the common era calender, the difference is still
very marked. We note that they are unlikely could be ascribed to the driving Earth system models (ESMs) owing to comparable

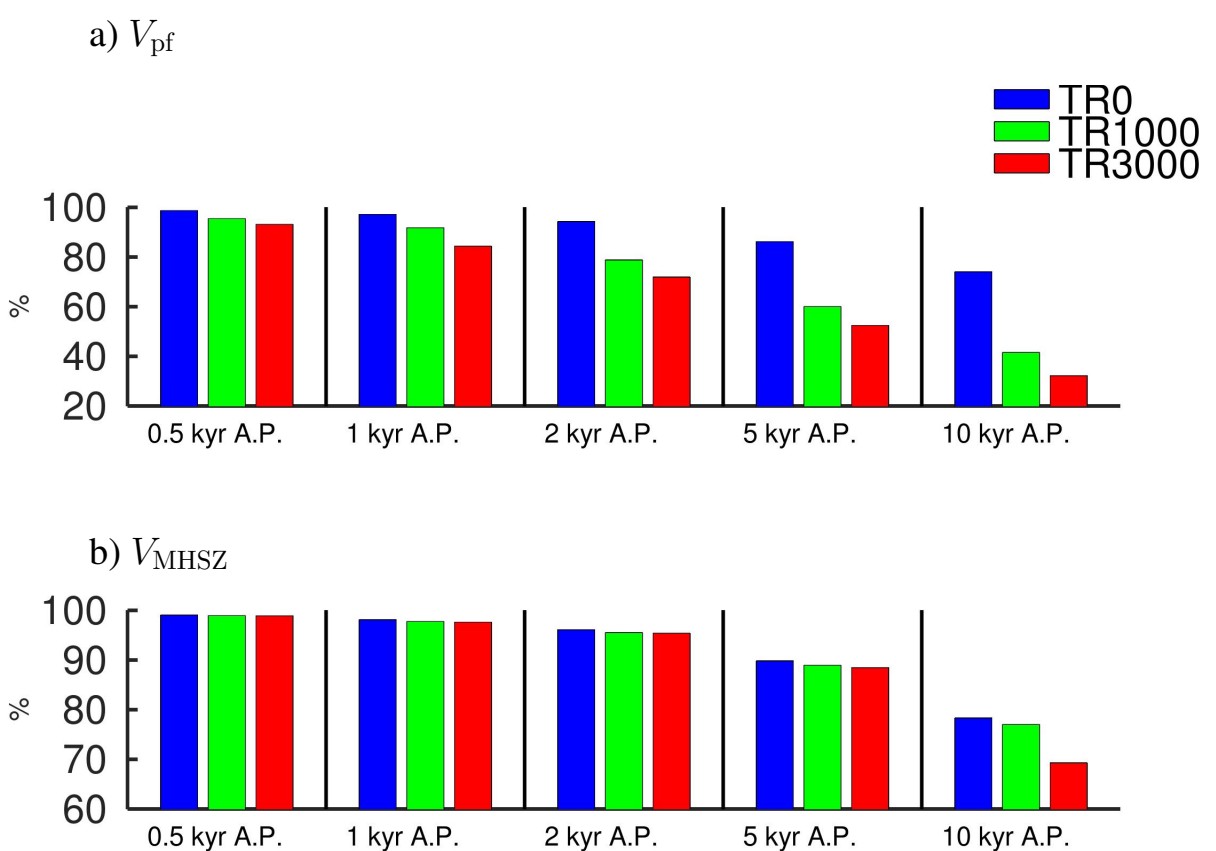

**Figure 6.** Volume of the subsea permafrost (a) and the permafrost-associated subsea methane hydrates stability zone (b) relative to the time instant of the $CO_2$ emission onset in simulations with $G = 60\,\mathrm{mW\,m^{-2}}$ averaged over the whole Arctic shelf.

equilibrium climate sensitivities (2.8 K) and transient climate responses (1.8 K) between MPI-ESM employed by Wilkenskjeld et al. (2021) and Climber-2 used in our exercise (MacDougall et al., 2020). The reasons behind difference in our simulations and in simulations reported by Wilkenskjeld et al. (2021) are unclear and devote further study.

After 10 kyr after the emission onset, in our simulations the subsea permafrost volume loss is, depending on $G$, is 16-44% in simulations TR0, 27-75% in simulations TR1000, and 37-86% in simulations TR3000. In this as well as in previous studied time slices the largest $V_\mathrm{pf}$ loss is exhibited for $G = 75\,\mathrm{mW\,m^{-2}}$ and the smallest one is for $G = 45\,\mathrm{mW\,m^{-2}}$.

In contrast to the subsea permafrost volume, the permafrost-associated MHSZ loss depends weaker on applied emission scenario (Fig. 6b) but strongly depends on $G$. This is again consistent with the finding that MHSZ mostly shrinks from below

rather than from above. The pan-Arctic methane hydrate stability zone volume, $V_\mathrm{MHSZ}$, is reduced by 0.4-3.4% during the first 0.5 kyr after the $CO_2$ emissions onset, by 1.0-7.8% during the next 0.5 kyr, by 2-16% to $t = 2\,\mathrm{kyr}$ A.P., by 4-45% to $t = 5\,\mathrm{kyr}$ A.P., and by 8-60% to $t = 10\,\mathrm{kyr}$ A.P. relative to its value at $t = 0$. Similar to that already exhibited for $V_\mathrm{pf}$, the



largest $V_{\mathrm{pf}}$ loss is exhibited for $G = 75 \, \mathrm{mW \, m^{-2}}$ and the smallest one is for $G = 45 \, \mathrm{mW \, m^{-2}}$. Our estimate of the relative $V_{\mathrm{MHSZ}}$ loss for $G = 60 \, \mathrm{mW \, m^{-2}}$ during the first 1 kyr after the emission onset is similar to that obtained by Hunter et al.

(2013) in their high-emission scenario ECP8.5 despite they do not model the permafrost-associated methane hydrates.

Geothermal heat flux is also very instrumental for setting the present-day simulated pan-Arctic methane stock, $M_{\mathrm{CH_4}}$, in the subsea hydrates. This stock is equal to $1230 \, \mathrm{PgCH_4}$ (all values for this variable are rounded to nearest integers) in simulations with $G = 60 \, \mathrm{mW \, m^{-2}}$ (Fig. 7a), but it is as half as much ($635 \, \mathrm{PgCH_4}$) in simulations with $G = 75 \, \mathrm{mW \, m^{-2}}$ and is larger by about 1/3 ($1695 \, \mathrm{PgCH_4}$) in simulations with $G = 45 \, \mathrm{mW \, m^{-2}}$. The majority of this stock (from 63% to 71% depending on $G$)

is in the East Siberian Arctic Shelf (ESAS), with smaller contributions from the West Eurasian and from the North American Arctic Shelves (17-20% and 11-17% correspondingly; for shelves definition see supplementary Fig. S3). The total pan-Arctic stock is broadly consistent with the synthesis by James et al. (2016) who figured out that up to $1400 \, \mathrm{PgCH_4}$ may be stored in the submerged permafrost in the Arctic shelf. However, our estimate is an order-of-magnitude larger that that reported by McGuire et al. (2009) ($\leq 65 \, \mathrm{PgCH_4}$). We note a strong dependence of the methane stock on $G$, quite similar to that obtained in

the present paper, was earlier simulated by Archer (2015) for ESAS, but with more moderate values of $M_{\mathrm{CH_4}}$ at this shelf (for instance, $846 \, \mathrm{PgCH_4}$ for $G = 60 \, \mathrm{mW \, m^{-2}}$, while his estimates are typically $\leq 90 \, \mathrm{PgCH_4}$), likely owing to his accounting for the $\mathrm{CH_4}$ dissolved content in the pore water.

Similar to that exhibited for MHSZ, the loss of the $M_{\mathrm{CH_4}}$ only weakly depends on the applied emission scenario during first several kiloyears after the emission onset (Fig. 7a), despite there is a strong corresponding dependence on $G$. By design, our

estimate of the relative methane hydrate stock decrease is identical to those obtained for the methane stability zone volume.

In contrast to MHSZ volume and its methane hydrate stock, the pan-Arctic methane flux from the sediments to the oceanic water, $F_{\mathrm{CH_4,w}}$, exhibits slightly non-monotonic dependence on $G$, For instance, for last 0.5 kyr before the emission onset, this flux is $1.91 \, \mathrm{TgCH_4 \, yr^{-1}}$ for $G = 60 \, \mathrm{mW \, m^{-2}}$ (Fig. 7b). It is slightly larger, $1.94 \, \mathrm{TgCH_4 \, yr^{-1}}$, for the largest studied $G = 75 \, \mathrm{mW \, m^{-2}}$, and is again larger, $2.40 \, \mathrm{TgCH_4 \, yr^{-1}}$, for the smallest studied $G = 45 \, \mathrm{mW \, m^{-2}}$. The major contribution

is from the East Siberian Arctic Shelf: for the last 0.5 kyr before the $\mathrm{CO_2}$ emission onset and for the first 0.5 kyr after this onset the flux from the sediments to the water from this shelf is from 1.3 to $3.4 \, \mathrm{TgCH_4 \, yr^{-1}}$ which is smaller than the estimate by Archer (2015), who reported that in his simulations $F_{\mathrm{CH_4,w}}$ from ESAS is $\leq 0.4 \, \mathrm{TgCH_4 \, yr^{-1}}$. The likely reason for this difference is due to i) an order of magnitude smaller methane stock in the ESAS sediments in his simulations compared to ours, and ii) accounting for future sea level rise on MHSZ extent (warming scenario in (Archer, 2015) corresponds to the sea

level rise $\geq 70 \, \mathrm{m}$ at the time of peak warming).

Future changes of $F_{\mathrm{CH_4,w}}$ depends on $G$ as well. If no $\mathrm{CO_2}$ emissions are applied (TR0), this flux steadily decrease until the complete MHSZ local extinction, when the final burst of the $\mathrm{CH_4}$ release takes place. In simulations with $\mathrm{CO_2}$ emissions applied, $F_{\mathrm{CH_4,w}}$ increases, sometimes by an order of magnitude. For instance, in simulation TR3000 with $G = 60 \, \mathrm{mW \, m^{-2}}$, even the being averaged over 0.5-1 kyr A.P., this flux is as large as $5.3 \, \mathrm{TgCH_4 \, yr^{-1}}$ (Fig. 7b); a higher counterpart value,

$6.6 \, \mathrm{TgCH_4 \, yr^{-1}}$, is obtained in simulation TR3000 with $G = 75 \, \mathrm{mW \, m^{-2}}$.





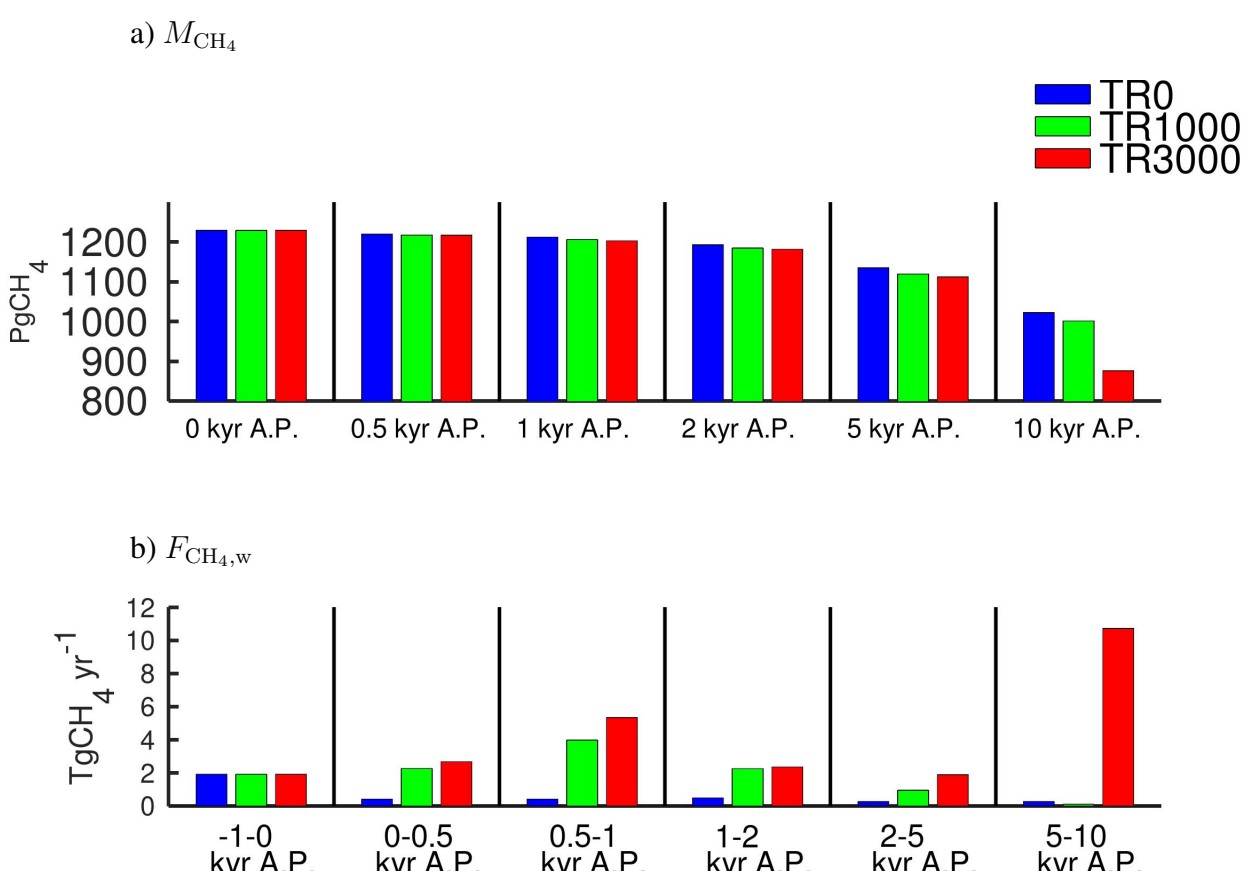

**Figure 7.** Methane stock in the pan-Arctic permafrost-associated shelf hydrate (a) and $CH_4$ flux from the Arctic shelf sediments to the oceanic water in simulations with $G = 60\,\mathrm{mW\,m^{-2}}$.

## 4 Discussion and conclusions

We performed simulations with the SMILES (the Sediment Model Invented for Long-tErm Simulations) for 100 kyr in future. This simulations were initialised from the state obtained for a broadly defined 'present-day state' and forced by surface air temperature (SAT) changes as simulated by the Climber-2 Earth system model. In turn, SAT changes in Climber-2 were

modelled as a response to idealised scenarios of $CO_2$ emissions and to changes of the parameters of the Earth orbit. Because of uncertainty in relating SAT changes to changes of temperature at the surface of the oceanic shelf sediments, $T_B$, we used a broad interval of future $T_B$ changes: from no change since the present day (which is a continuation of the Holocene history and could be an underestimate at the shallowest part of the shelf) to the change with same rate as it is simulated for SAT (which is apparently a drastic overestimate out of this shallowest part). Owing to additional uncertainty due to existing spatial

variations of geothermal heat flux, we repeated model runs for several values of this variable ranging from 45 to $75\,\mathrm{mW\,m^{-2}}$. We neglected possible impact of the sea level on freeze/thaw temperature and on thermodynamic stability of methane hydrates.





We found that for the outer shelf ($H_{\mathrm{D}} = 100\,\mathrm{m}$) permafrost is either disappears before $t = 0$ or is simulated to disappear during few centuries in future provided that $G \geq 60\,\mathrm{mW\,m^{-2}}$. For smaller $G$ and at the same part of the shelf, the date of the complete degradation is not later than 11 kyr A.P. depending on the applied emission scenario. For the middle and shallow parts of the shelf, in the $CO_2$-emission forced runs the subsea permafrost survives, at least, for 5 kyr after the emission onset or even much longer. Without applied greenhouse forcing, the permafrost exists here at least until 22 kyr .A.P. or even survives till the end of the model runs.

At the shallow and middle parts of the shelf in our simulations methane hydrate stability zone disappears not earlier that at $t = 3\,\mathrm{kyr\,A.P.}$, but typically MHSZ survives until 11 to 20 kyr A.P. (from 31 to 41 kyr A.P.) for $G = 60\,\mathrm{mW\,m^{-2}}$ and $H_{\mathrm{D}} = 50\,\mathrm{m}$ ($H_{\mathrm{D}} = 10\,\mathrm{m}$). For smaller geothermal heat flux $G = 45\,\mathrm{mW\,m^{-2}}$, the time instant of the local MHSZ extinction is from 36 to 81 kyr A.P. depending on the $CO_2$ scenario and on the contemporary shelf depth.

Timings of local extinction of both the subsea permafrost and MHSZ are negatively correlated with the geothermal heat intensity provided that other factors are being equal. This reflects the strong control which this variable sets on the permafrost thaw from the bottom and the corresponding MHSZ shrinking. In turn, thaw from the top and MHSZ table deepening are basically determined by the applied $CO_2$ forcing scenario. Because of different contribution of processes at the top and bottom boundaries to total loss for these two variables, the time instants of the permafrost disappearance depend stronger on the applied scenario relative to that of the MHSZ extinction.

Despite the simplistic setup of our experiments (in particular, a very rudimental treatment of geographical variations of all governing parameters), we attempted to make order-of-magnitude pan-Arctic estimates for properties of the subsea permafrost and the permafrost-associated methane hydrates. For instance, the present-day pan-Arctic methane stock in the subsea hydrates is from 635 to 1695 $\mathrm{PgCH_4}$ depending on $G$ with a major contribution ($\approx 2/3$) from the East Siberian Arctic Shelf.

During the first 0.5 kyr centuries after the $CO_2$ emissions onset, subsea permafrost volume is decreased by up 2% under scenario of fixed temperature at the top of the sediments, and by up to 13% under high-emission scenario. At $t = 1\,\mathrm{kyr\,A.P.}$, this volume in simulations TR0 is decreased, respectively, by up 5% and by up 20% (all values are relative to the time of the $CO_2$ emission onset). After 10 kyr after the emission onset, the corresponding loss is up to 44% and up to 86%.

The permafrost-associated MHSZ loss is more moderate and depends weaker on applied emission scenario. The pan-Arctic methane hydrate stability zone volume diminishes by up to 3% during the first 0.5 kyr after the $CO_2$ emissions onset, by up to 8% during the next 0.5 kyr, and by up to 60% at $t = 10\,\mathrm{kyr\,A.P.}$.

We conclude that the $CO_2$-induced warming in our simulations enhances the pan-Arctic subsea permafrost loss during 1 kyr after the emissions onset by several times. However, this warming is much less instrumental for the respective MHSZ loss.

Our estimates of methane release from the sediments to the oceanic water per unit area are consistent with existing respective estimates attributed to the subsea permafrost thaw. Our corresponding pan-Arctic estimate averaged over centennial or millennium time intervals never exceeds several $\mathrm{TgCH_4\,yr^{-1}}$ though except for periods when MHSZ is near ceasing to exist and $CH_4$, which was accumulated beneath it during earlier MHSZ shrinking, is instantaneously released from the sediments.

We did not make an attempt to estimate the corresponding release of methane from the oceanic water to the atmosphere because this is complicated by i) methane oxidation in the water column, ii) sea ice blocking of gas transport from the ocean to





the atmosphere, and iii) additional $CH_4$ sources unrelated to the methane hydrate dissociation, e.g., due to methanogenesis in the river mouths or in the submerged yedoma or owing to release of the thermogenic methane, (Archer, 2007; James et al., 2016; Ruppel and Kessler, 2017; Dean et al., 2018; Shakhova et al., 2019; Ruppel and Waite, 2020). Assessing item iii is beyond the

scope of the present paper. For items i and ii, it is clear that they could only diminish $CH_4$ flux at the ocean-atmosphere interface relative to its counterpart at the sediment-ocean interface. According to (Malakhova and Golubeva, 2021), the difference may be as large as one order of magnitude provided that sea ice cover is similar to the present-day one. While the latter assumption seems unlikely for time instants after several decades from the present day, when most state-of-the-art climate models projects ice-free Arctic in summer under high emission scenarios (Fox-Kemper et al., 2021), a severalfold decrease of methane flux

at the ocean-atmosphere interface relative to the flux at the sediment-ocean interface still seems reasonable. We suggest that $CH_4$ flux at the ocean-atmosphere interface could not be larger than few $TgCH_4 \, yr^{-1}$. Thus, despite our pan-Arctic $F_{CH_4,w}$ is substantially larger than estimated by Archer (2015), we agree with his conclusion that methane hydrate dissociation in the subsea sediments can not support large estimates which were claimed recently (e.g., up to $17 \, TgCH_4 \, yr^{-1}$, Shakhova et al., 2010). The same conclusion was made in the recent IPCC assessment (Canadell et al., 2021) (see also Berchet et al., 2016).

We acknowledge the limitations of our study:

- In our simulations, we neglected impact of future sea level (SL) rise on thermodynamic stability of methane hydrates and on freeze/thaw temperature of sea water. For both effects, this is true provided that the eustatic sea level rises only due to thermal expansion. However, it overlooks possible contribution to the sea level rise due to melting of ice sheets. For instance, the recent IPCC Working Group 1 Assessment Report concluded that ice sheets melting may contribute

as large as 40 m to sea level within few centuries from now (Fox-Kemper et al., 2021). Nonetheless, ice sheets do not melt in future in the Climber-2 simulations used here, and their contribution to the future SL rise is zero. In addition, if ice sheet melting to SL change is substantial, it would only enhance thermodynamic stability of methane hydrates, thus, shifting the dates of MHSZ extinction even further in future. However, the hydrostatic pressure increase would decrease freeze/thaw temperature (supplementary Eq. (S5)), which would promote a faster response of the subsea permafrost to

future warming.

- An important caveat is due to lack of mechanistic biogeochemistry in our model reflected in vertically and geographically uniform $\theta_{CH_4}$. This simplification may affect our results in two ways. First, it is directly affects the estimated methane release. Second, it overlooks a negative feedback between the dissociation of methane hydrates and the amount of dissolved methane in pore volume (Ruppel and Waite, 2020). The first effect is difficult to quantify in our setup. The

second effect may only slow sown the hydrate dissociation, thus, diminishing the release of methane and shifting the extinction of methane hydrates in the sediments further in time.

- We used very idealised scenarios of $CO_2$ emissions leading to the 'calender uncertainty' in our simulations. In addition, other external forcings are neglected completely except the orbital one. We note, however, that long time scales involved in the problem at hand suppress possible impact of scenario details on the obtained results. At least, the major finding

related to the major dependence of the time instants of complete extinction to the shelf depth and to the processes at



the bottom of the permafrost layer and of MHSZ have to be valid irrespective to the applied scenario emission. Further, at long scale, the climate response is determined basically by the cumulative emissions rather than by the pathway details (Zickfeld et al., 2009, 2012). This also provides are support that our estimates are correct at least to the order of magnitude.

– An obvious limitation is due to our selection of the particular Climber-2 grid cell for climate anomalies and the spatially uniform present-day temperature to which these anomalies are added, both are corresponding to the East Siberian Arctic shelf (supplementary Sect. S3). This is partly reasoned by the major contribution to the permafrost area, MHSZ volume, and MHSZ stock from this part of the Arctic shelf as well as by relatively uniform projections of temperature at these latitudes in Climber-2. In addition, it directly affects only the present day state rather than future simulations.

– We used hydrostatic pressure to calculate MHSZ boundaries (Sect. S1). This is similar to (Romanovskii et al., 2005; Majorowicz et al., 2012; Hunter et al., 2013; Archer, 2015) and is equivalent to assuming that part of the column remains unfrozen even at very low temperature. However, in some papers (e.g., Tinivella and Giustiniani, 2013; Portnov et al., 2014; Liu et al., 2016) alternative pressure calculations, which take into account the lithostatic pressure, are invoked. The latter presumes an existence of completely hydrologically impermeable layers in the sediments and leads to a
deeper MHSZ base (Tinivella and Giustiniani, 2013). The impact of the the replacement of the hydrostatic pressure by a lithostatic one is not explored in our paper.

– We reported the pan-Arctic estimates only for $G = 60\,\mathrm{mW\,m^{-2}}$. This is a typical heat flux from the Earth interior in the Arctic Ocean (Davies, 2013). Arctic shelf regions with substantially larger $G$ loose MHSZ markedly before the emission onset and do not contribute to the estimates. The regions with much smaller geothermal heat flux are untypical for the
Arctic shelf.

– At last, there is a caveat in our simulation for $\left(H_\mathrm{D} = 10\,\mathrm{m}; G = 45\,\mathrm{mW\,m^{-2}}\right)$ with MHSZ extending down to the bottom of the computation domain boundary and the bottom of the permafrost layer located close to the bottom of the computation domain. This would apparently lead to the misestimated values of all variables under interest. We acknowledge it and we are going to ameliorate it in future exercises. However, this pair $(H_\mathrm{D}; G)$ does not look like
'an outlier' in our simulations, and we believe that the results for this pair are correct at least qualitatively. In addition, the extent of areas with this geothermal heat flux in the Arctic is small (Davies, 2013), and all pan-Arctic estimates are done for more common $G = 60\,\mathrm{mW\,m^{-2}}$.

Therefore, we conclude that our estimates, albeit may be improved in a more detailed setup, are still correct up to the order of magnitude.

*Code and data availability.* The SMILES simulations output used in this paper is available at the ZENODO repository via https://doi.org/10.5281/zenodo.5728529. The SMILES code is available from the first author by request.





*Author contributions.* Both authors conceptualised the study. V.V.M. performed model simulations. Both authors contributed to the manuscript preparation.

*Competing interests.* The authors declare that they have no conflict of interest.

*Acknowledgements.* Authors are grateful to A. Ganopolski for making available the Climber-2 output. The work is supported by the Russian Science Foundation grant 21-17-00012.



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





## List of Figures