# Peer review of "Subsea permafrost and associated methane hydrates: how long will they survive in the future?"

_Earth System Dynamics, 2021_

## Referee Comment (RC1)

**Review of "Subsea permafrost and associated methane hydrates: how long will they survive in the future?" by V.V.Malakhova and A.V.Eliseev, in Earth System Dynamics Discussions, esd-2021-99**

Stiig Wilkenskjeld

March 3, 2022

In this interesting manuscript, the authors present the development of the subsea permafrost and the methane hydrate stability zone up to 100.000 years into the future simulated by a 1D-model for points representing different points on the East Siberian Shelf. They perform scenario simulations using different climate projections in combination with different assumptions about the geothermal heat flux. The manuscript assesses the very relevant question on the stability of the Arctic subsea permafrost and its role in the climate system using a set of interesting experiments.

Abbreviations used in this review: SSPF = SubSea PermaFrost, MHSZ = Methane Hydrate Stability Zone.

**Major comments**

Both in the abstract and in the discussion/conclusion many numbers from the results are stated. However, I would (in both places) like to read one or two sentences on the main conclusions/the "take-home-message(s)" of the paper. My personal favourite is, that according to this study, MHSZ development is independent on the chosen climate projection, at least for several thousand years.

At several places, it is mentioned that this study (in contrast to earlier studies, e.g. Archer [2015]), the changes in the orbital parameters of the Earth are taken into account. It is however nowhere discussed which influence this has on the results.

The upscaling to pan-Arctic scale (Sec. 3.4) is — as it is also clearly stated in the manuscript — somewhat speculative due to the many assumptions needed for the upscaling. It could be considered part of the discussion instead of as "a" result. This specifically holds for the comparisions to other studies (e.g.

Wilkenskjeld et al. [2021], lines 274–285).

Also a part of the model description (line 128–133, comparing the setup to Archer [2015]) could advantageously be postponed to the discussion.

**Minor comments**

That the geography is in the model represented by "representative points" should be more emphasized — specifically also in the abstract.

The vertical setup (and thus type) of the model is needed in the model description. I.e. that it's a discrete $0.5\,m$ vertical grid down to $1500\,m$.

Much of the model description is found both in the manuscript and the supplement. The supplement could be shortend.

Line 78: "a condition of temperature continuity". How is continuity defined on a discrete grid?

Figure 1: The general shape of the figures is intuitive, however some features seems rather pecuilar:

1. Some very steep deepening (from top)/rising (from bottom) is present, most obvious in $H_D = 10\ m, G = 45mW\,m^2$ for TR1000/TR3000. Expected is more a shape like TR3000 in $H_D = 50\ m, G = 45mW\,m^2$

2. The wave-like structure on the lower boundary, mainly visible in $H_D = 100\ m, G = 45mW\,m^2$.

Comments on these features would be appreciated.

Figure 1: I would also show the panel on $H_D = 100\ m, G = 75mW\,m^2$ even though it's empty. It would save many explanations, and the space for the panel is anyway available.

Figure 1: Consider using the same Y-axis for every subplot in a row.

Line 189: As I read the figures, MHSZ never extends above $200\,m$ (Fig. 2) depth whereas SSPF is present near the surface at $t = 0$ (Fig. 1). This seems to contradict the sentence here.

Line 198: Which simulations are meant by "simulations with shallowing rate of $v_{\mathrm{MHSZ,b}}$"? Meaning is here not clear.

Line 216-218: Would it not be more realistic to assume that (also) SSPF prevents methane from escaping the sediments? In this way the methane puls will only escape when both MHSZ and SSPF is gone.

Line 216-218: How would the methane flux to the ocean develop without this assumption? Of course it is reasonable to argue that SSPF and MHSZ acts as a lid preventing outgasing. However, it is likely that this lid is not completely closed (due to cracks and other geological features), and thus it would provide an interesting upper-limit to the methane fluxes in the relatively near future to look at the results without this assumption.

Line 222: Should be "sediment-to-ocean" rather than "ocean-to-atmosphere"? (Since the chemical fate of the methane in the ocean water colume is nowhere quantified.)

Line 238-239: I don't understand how an order-of-magnitude difference can arrise as a consequence of a factor-6 difference in averaging length of a quantity given as a flux.

Figure 5b: The Y-scale make the results hardly readable. Better would be to let the extreme values ($G = 75\ mW\ m^2, 2-5\ kyr$ and evt. $G = 60\ mW\ m^2, 5-10\ kyr$, TR3000) go off-scale (values stated in the figure caption) and plot only $Y = 0..10\ g\ m^{-2}\ yr^{-1}$ (as in subfigure a).

The lines 274–284 are devoted to a comparison to my (and co-autor's) study (Wilkenskjeld et al. [2021]), where the authors speculate on the big differences between our results. I guess the most important reason for the differences is our use of "partially frozen cells", an approach partly inherited from the SuPerMAP model [Overduin et al., 2019] delivering our inititial conditions and partly necessary due to our rather coase resolution horizontally and in-depth also vertically. Though the initial conditions of the present and our study roughly agree on the location of tbe bottom of the SSPF, the present study likely have a much large volume of deep (below $100\ m$) SSPF ice (Fig. 1, see also Fig. 1b in Wilkenskjeld et al. [2021]). This ice is not affected by climate within the next 1000 years, and therefore we, by thawing the upper ice away, have be thawing a much larger fraction of the total SSPF ice, eventhough the two studies likely thaw similar amounts of ice.

Line 297-299: The numbers presented for methane captured in the MHSZ are huge compared to any to me known estimated. Also it is not very clear where these numbers come from. Is it due to the assumption that the MHSZ is completely saturated? If "yes": is this assumption realistic?

Line 316-317: As I read this sentence, it is clamed that 1.3 (or 3.4) is less than 0.4?

Line 358: "scenario of fixed temperature": Guess this means "TR0", which would be more readable.

In many cases of the bar charts (Fig. 5-7), I could imagine that the message would be clearer by using (properly smoothed) time series — eventually with non-linear time axes. This is of course a very personal opinion.

Not so much for the manuscript, but rather for my personal curriosity: Is any statement possible on the influence of salinity diffusion (which was not included in my own study)?

[Figure]

Figure 1: *Area of SSPF ice as function of depth for the initial conditions (solid lines) and at end-of-experiment for the SSP5-8.5 scenario (dashed lines) in Wilkenskjeld et al. [2021]. Different interpretations of the data are shown in different colors: Black: fractions applied (as used in Wilkenskjeld et al. [2021], red: any ice present is interpreted as cell full of ice, blue: more than 50% ice is interpreted as cell full of ice (less than 50% means no ice). The insert shows the same thing but at a logarithmic depth (Y) scale. The alternative interpretations (blue and red) of the scenario data (dashed) can of course only be interpreted as a momentary state and not as a simulation result based on the initial state, since here much more ice has disapeared than in the actual simulations. They are shown to illustrate that very little happens in the depth, also using the alternative interpretations.*

**Language, presentation and technical comments**

The language of the manuscript is with few exceptions fully understandable. However, the readability could be in many cases be improved, presumably by correction by an englisch native speaker.

In many cases an additional word (often conjugations of "to be") is present in a sentense. This could either be leftovers of previous versions of the sentences or some general language differences between russian and english.

Line 2: "Earth System Model" (all with initial capitals).

In section 3.3 (specifically from Eq. (2)) the term $f_{CH_4}$ is used, later on and in the figures $F_{CH_4}$ is used. Please choose one of the versions.

Equation 1: The factor $\phi$ is either there by accident or not described in the text.

Line 232: Repetition of "TRx000" unnecessary.

Line 234: Guess the meaning is "ceases to exist" (not "exit").

Line 357: "0.5 kyr centuries" seems to be a mixture of two sentence versions.

Line 376: Reference style error (wrong bracket placement).

Line 400: "sown" = "down"?

**References**

D. Archer. A model of the methane cycle, permafrost, and hydrology of the siberian continental margin. *Biogeosciences*, 12 (10):2953–2974, 2015. doi: 10.5194/bg-12-2953-2015. URL https://bg.copernicus.org/articles/12/2953/2015/.

P. P. Overduin, T. Schneider von Deimling, F. Miesner, M. N. Grigoriev, C. Ruppel, A. Vasiliev, H. Lantuit, B. Juhls, and S. Westermann. Submarine permafrost map in the arctic modeled using 1-d transient heat flux (supermap). *Journal of Geophysical Research: Oceans*, 0(0), 2019. doi: 10.1029/2018JC014675. URL https://agupubs.onlinelibrary.wiley.com/doi/abs/10.1029/2018JC014675.

S. Wilkenskjeld, F. Miesner, P. P. Overduin, M. Puglini, and V. Brovkin. Strong increase of thawing of subsea permafrost in the 22nd century caused by anthropogenic climate change. *The Cryosphere Discussions*, 2021:1–18, 2021. doi: 10.5194/tc-2021-231. URL https://tc.copernicus.org/preprints/tc-2021-231/.

---

## Author Comment (AC1)

**Reply to the reviewer's comments to**
**Subsea permafrost and associated methane hydrates: how long will they survive in the future?**

V.V. Malakhova and A.V. Eliseev

June 20, 2022

We are grateful for the reviewer for the constructive and insightful comments which led to the improved presentation of our results.

The most important changes in the manuscript are as follows:

- Supplementary information is extended by figures showing

    - profiles of temperature and salinity at $t = 0$;

    - $T_B$ before $t = 0$;

    - results of the ACCESS ESM-1.5 SSP5-8.5 simulation for seafloor temperature in support of our choice for future scenarios of climate change;

    - permafrost layer and MHSZ simulation from 400 kyr B.P. to 0 kyr B.P.

- We dropped out the assumption that MHSZ is an impermeable layer for $CH_4$ transport. As it was expected, this resulted in a larger methane flux at the sediment-ocean interface during the gradual MHSZ degradation and eliminated the pulse release of methane at the end of this process. However, we still discuss the respective results from the previous simulations (because it is a potentially interesting sensitivity study) The former Fig. 5 is moved into the Supplement.

- Some figures are redrawn and restructured. In particular, the former Figs. 2 and 4 and combined into a single Figure (now referred to as Fig. 2). This is done to make it easier to compare the time of disappearance for permafrost and for MHSZ. Other figures are renumbered accordingly.

- Upon revising our paper, we found an error in our calculations: flux $f_{CH_4}$ *was not* multiplied by $K_S$ except that belonging to the pan-Arctic estimates. Now, this error is corrected.

- We found a logical inconsistency in our notations. In particular, the present-day shelf depth (thickness of water layer above the sea floor) was denoted as $H_\mathrm{D}$, while temperature at the sediment water interface was referred to as $T_\mathrm{B}$. Now, $H_\mathrm{D}$ is replaced by $H_\mathrm{B}$ to highlight that they point to the characteristics at the same physical surface.

Below, the point-to-point replies to the comments are presented. The original comments are typed in italics, and the replies are typed in regular font.

**General comments**

- *Both in the abstract and in the discussion/conclusion many numbers from the results are stated. However, I would (in both places) like to read one or two sentences on the main conclusions/the "take-home-message(s)" of the paper. My personal favourite is, that according to this study, MHSZ development is independent on the chosen climate projection, at least for several thousand years.*
  Yes, we agree that abstract should be shortened. Upon revision, some numbers are removed, and some statements are revised. In particular, it is stated that MHSZ dynamics is independent on the chosen climate projection, at least for the next several thousand years.

- *At several places, it is mentioned that this study (in contrast to earlier studies, e.g. Archer [2015]), the changes in the orbital parameters of the Earth are taken into account. It is however nowhere discussed which influence this has on the results.*
  The most important impact of the future orbital forcing is a non-monotonic change of $T_\mathrm{B}$. It impact on simulations is more important for the permafrost than for MHSZ. In particular, it leads to retardation of the permafrost table thaw rate in TR1000 and TR3000. However, its effect is not a dominant one, because such thaw (albeit with a much reduced rate) is exhibited in simulation TR0 as well.

- *The upscaling to pan-Arctic scale (Sec. 3.4) is – as it is also clearly stated in the manuscript – somewhat speculative due to the many assumptions needed for the upscaling. It could be considered part of the discussion instead of as "a" result. This specifically holds for the comparisons to other studies (e.g. Wilkenskjeld et al. [2021], lines 274–285).*
  We agree that this upscaling is rather speculative. Thus, we moved this material to the 'Conclusions and Discussion' section. The latter is now subdivided into three subsections to simplify reading.

- *Also a part of the model description (line 128–133, comparing the setup to Archer [2015]) could advantageously be postponed to the discussion.*
  This paragraph is moved to Sect. 2.

**Specific comments**

- *That the geography is in the model represented by "representative points" should be more emphasized – specifically also in the abstract.*
  This clarification is added to Sect. 2 of the manuscript.

- *The vertical setup (and thus type) of the model is needed in the model description. I.e. that it's a discrete 0.5 m vertical grid down to 1500 m.*
  An information on the vertical grid is added to Sect. 2 of the main text.

- *Much of the model description is found both in the manuscript and the supplement. The supplement could be shortened.*
  We prefer to keep the model description in the Supplementary Information as detailed as possible. Otherwise, it would be quite tedious for a reader to merge different pieces of information from the main text and from the Supplement.

- *Line 78: "a condition of temperature continuity". How is continuity defined on a discrete grid?*
  We agree that the wording is awkward. Temperature continuity is a step in derivation of the Stefan condition. Namely, it is assumed that temperature is the same just above and just below thaw freezing/thaw interface. Thus, the sentence on temperature continuity is removed from the manuscript, and the only note on Stefan condition is kept in the text.

- *Figure 1: The general shape of the figures is intuitive, however some features seems rather peculiar:*
  *1. Some very steep deepening (from top)/rising (from bottom) is present, most obvious in $H_D = 10$ m, $G = 45$ mW $m^{-2}$ for TR1000/TR3000. Expected is more a shape like TR3000 in $H_D = 50$ m, $G = 45$ mW $m^{-2}$.*
  *2. The wave-like structure on the lower boundary, mainly visible in $H_D = 100$ m, $G = 45$ mW $m^{-2}$.*
  *Comments on these features would be appreciated.*
  Yes, thank you for pointing this out. The comments are as follows:
  1. Fast (but at the multi-millennium timescale) thaw from the top is a continuation of the thaw induced by the last glacial termination. The bottom thaw rate is always between 10 and 20 m kyr$^{-1}$ in our simulations. Fast thaw from above is exhibited only in simulations which are forced by anthropogenic emissions. These emissions result in increase of the permafrost table thaw rate from $\approx 1.5$ m kyr$^{-1}$ (TR0) to 13 m kyr$^{-1}$ (TR3000). Both conclusios are in the text already.
  2. Wavy structure is a combination of two phenomena. The first one is an impact of isothermic thaw of pore ice leading to stop of the thaw front movement when heat is accumulated, and a renewal of such movement when the accumulated heat is enough to melt the pore ice at a discrete vertical grid. The second one (playing the more important role at the bottom of the permafrost) is a coarse-scale output of our model – we store

the data once per 100 yr. Taking into account the mentioned-above thaw rates, such coarse-scale output results in the movement of the thaw interface by less than one grid step during a single time step, thus, leading to the 'jumps' when such interface suddenly changes position between the nearby grid cell. Upon revision, boundaries in Fig. 1 are smoothed with a window length of 1 kyr to remove the mentioned-above 'jumps'. The respective note is added to the caption of the Figure.

- *Figure 1: I would also show the panel on $H_D = 100$ m, $G = 75$ mW $m^{-2}$ even though it's empty. It would save many explanations, and the space for the panel is anyway available.*
  When this figure is redrawn with the subplot-independent $Y$-ranges (see the next comment; the same comment was due to the another reviewer as well), panels for $H_{\mathrm{B}} = 100$ m look very non-informative. Therefore we chose to remove these panels from figure entirely.

- *Figure 1: Consider using the same $Y$-axis for every subplot in a row.*
  Upon revision, this figure is redrawn with $Y$-range from 0 to 1500 m.

- *Line 189: As I read the figures, MHSZ never extends above 200 m (Fig. 2) depth whereas SSPF is present near the surface at $t = 0$ (Fig. 1). This seems to contradict the sentence here.*
  Yes, we agree. 'Smaller' is replaced by 'larger'.

- *Line 198: Which simulations are meant by "simulations with shallowing rate of $v_{\mathrm{MHSZ,b}}$"? Meaning is here not clear.*
  It was a misprint. A correct sentence reads, 'In the simulations with other values of $G$, the rate of $v_{\mathrm{MHSZ,b}}$ averaged over 5 kyr A.P. $\leq t \leq$ 10 kyr A.P. is close to 100 m kyr$^{-1}$ for all three emission scenarios'.

- *Line 216-218: Would it not be more realistic to assume that (also) SSPF prevents methane from escaping the sediments? In this way the methane pulse will only escape when both MHSZ and SSPF is gone.*
  This is potentially interesting, but it would increase the volume of our paper dramatically. We would plan to do this in future.

- *Line 216-218: How would the methane flux to the ocean develop without this assumption? Of course it is reasonable to argue that SSPF and MHSZ acts as a lid preventing outgasing. However, it is likely that this lid is not completely closed (due to cracks and other geological features), and thus it would provide an interesting upper-limit to the methane fluxes in the relatively near future to look at the results without this assumption.*
  Yes, we agree. In our new calculations, MHSZ is permeable for a methane transport. This removed the pulse release of methane at the time of the complete extinction of MHSZ and increased methane fluxes during gradual degradation of MHSZ. Old calculations are moved to the Discussion section, and the former Fig. 5 is moved to the Supplement. Supplement. In addition, our estimate for methane fluxes was constructed assuming an

instantaneous transport of methane from MSHZ to the sea floor (Sect. 2). In reality, this transport is controlled by diffusion and vertical advection. Both processes result in a finite timescale for such transport (Xu, Ruppel, 1999). Therefore, it is likely that our assumption of an instantaneous transport of methane leads to the overestimated corresponding flux at the sediment-ocean interface. The respective paragraph is added to Sect. 4.3.

- *Line 222: Should be "sediment-to-ocean" rather than "ocean-to-atmosphere"? (Since the chemical fate of the methane in the ocean water column is nowhere quantified.)*
  Yes, thanks. The sentence is corrected.

- *Line 238-239: I don't understand how an order-of-magnitude difference can arise as a consequence of a factor-6 difference in averaging length of a quantity given as a flux.*
  We meant that different averaging intervals may either include or not include the pulse release at the timing of the MHSZ disappearance. However, we agree that this sentence is unclear and might be misleading. Moreover, pulse release is not exhibited in our new set up. It is excluded upon revision.

- *Figure 5b: The Y-scale make the results hardly readable. Better would be to let the extreme values ($G = 75$ $mW$ $m^{-2}$, 2-5 kyr and evt. $G = 60$ $mW$ $m^{-2}$ , 5-10 kyr, TR3000) go off-scale (values stated in the figure caption) and plot only $Y = 0 \ldots 10$ $g$ $m^{-2}$ $yr^{-1}$ (as in subfigure a).*
  Because the assumption of MHSZ impermeability is dropped in the revised manuscript, there is no need to this construction of the figure. The revised figure (which is Fig. 4 now owing to merging previous Figs. 2 and 4 into a single figure) is drawn in a more spectacular way.

- *The lines 274–284 are devoted to a comparison to my (and co-author's) study (Wilkenskjeld et al. [2021]), where the authors speculate on the big differences between our results. I guess the most important reason for the differences is our use of "partially frozen cells", an approach partly inherited from the SuPerMAP model [Overduin et al., 2019] delivering our initial conditions and partly necessary due to our rather coarse resolution horizontally and in-depth also vertically. Though the initial conditions of the present and our study roughly agree on the location of the bottom of the SSPF, the present study likely have a much large volume of deep (below 100 m) SSPF ice (Fig. 1, see also Fig. 1b in Wilkenskjeld et al. [2021]). This ice is not affected by climate within the next 1000 years, and therefore we, by thawing the upper ice away, have be thawing a much larger fraction of the total SSPF ice, even though the two studies likely thaw similar amounts of ice.*
  We are very grateful for this insightful comment. The respective note is added to Sect. 4.2.

- *Line 297-299: The numbers presented for methane captured in the MHSZ are huge compared to any to me known estimated. Also it is not very clear where these numbers come from. Is it due to the assumption that the MHSZ is completely saturated? If "yes": is this assumption realistic?*
  We do not assume that hydrates are completely saturated. It is stated in the paragraph right after Eq. (1) that our assumed saturation is $\theta_{CH_4} = 0.05$. However, we acknowledge that our value (1230 PgCH$_4$) is an overestimate. The likely reasons for obtaining such value are i) the assumption that hydrates exist everywhere in the MHSZ, while Xu and Ruppel (1999) and Mestdagh et al. (2017) pointed out the hydrates are mostly absent in the uppermost part of MHSZ, and they do exist in the lowermost part only provided that CH$_4$ flux from below is large enough, ii) possible unfrozen (and, thus, unable to support the thermodynamic conditions for hydrate formation) horizontal subgrid-scale regions. Both assumptions might lead to the several-fold overestimated methane stock, and in combination they might lead to the corresponding overestimate by order of magnitude. The respective discussion is added to Sect. 4.2.
  Nonetheless, quite a similar value (1400 PgCH$_4$) was reported by James et al. (2014) as based on Shakhova et al. (2010). We this estimate to new Fig. 7 for clarity.
  In addition, it is important that even our (likely overestimated) methane stock in the Arctic shelf sediments is unable to support large methane fluxes which are reported, for instance, by Shakhova et al. (2010). The respective note is added to the subsection on pan-Arctic estimates.

- *Line 316-317: As I read this sentence, it is claimed that 1.3 (or 3.4) is less than 0.4?*
  Sorry for this misprint. It should be 'larger' rather than 'smaller'.

- *Line 358: "scenario of fixed temperature": Guess this means "TR0", which would be more readable.*
  Upon revision, the entire paragraph is removed from the manuscript.

- *In many cases of the bar charts (Fig. 5-7), I could imagine that the message would be clearer by using (properly smoothed) time series — eventually with non-linear time axes. This is of course a very personal opinion.*
  We tried this option many times during manuscript preparation and revision. This was always less readable than our bar charts, especially for the permafrost and MHSZ (which dynamics are the major goal of our paper) because of the necessity to put 9 (3 cases for $G$ and 3 cases for $H_B$) on the same plot.

- *Not so much for the manuscript, but rather for my personal curiosity: Is any statement possible on the influence of salinity diffusion (which was not included in my own study)?*
  In our previous manuscript (Malakhova and Eliseev, 2020b) it was found that the impact of salinity diffusion on the permafrost-associated methane

hydrates is not marked due to deep level of their occurrence in the shelf sediments. While this is not a strong conclusion, we prefer not to go deeper in this matter at the date, because to arrive at a firm conclusion require to set up specific simulations.

**Language, presentation and technical comments**

- *In many cases an additional word (often conjugations of "to be") is present in a sentense. This could either be leftovers of previous versions of the sentences or some general language differences between russian and english.*
  The language is checked and ameliorated.

- *Line 2: "Earth System Model" (all with initial capitals).*
  The sentence is revised accordingly.

- *In section 3.3 (specifically from Eq. (2)) the term $f_{CH_4}$ is used, later on and in the figures $F_{CH_4}$ is used. Please choose one of the versions.*
  Upon revision, we clarified our terms. We use $f$ for fluxes per unit area (mass per unit area per unit time) and $F$ for the area-summed fluxes (mass per unit time). We agree that these letters were used in a somewhat confusing way in our previous manuscript version. Now this is ameliorated. In addition, a note is added on the difference between $f$ and $F$ as well as on the difference between $m$ and $M$.

- *Equation 1: The factor $\phi$ is either there by accident or not described in the text.*
  This is porosity. It is defined earlier, in a brief description of SMILES.

- *Line 232: Repetition of "TRx000" unnecessary.*
  Now this repetition is replaced 'with external $CO_2$ emissions'.

- *Line 234: Guess the meaning is "ceases to exist" (not "exit").*
  The misprint is corrected.

- *Line 357: "0.5 kyr centuries" seems to be a mixture of two sentence versions.*
  Upon revision, this sentence is removed from the paper.

- *Line 376: Reference style error (wrong bracket placement).*
  The sentence is revised accordingly.

- *Line 400: "sown" = "down"?*
  The misprint is corrected.

---

## Author Comment (AC2)

*Reply to the reviewer's comments to*
Subsea permafrost and associated methane hydrates: how long will they survive in the future?

V.V. Malakhova and A.V. Eliseev

June 20, 2022

We are grateful for the reviewer for the constructive and insightful comments which led to the improved presentation of our results.

The most important changes in the manuscript are as follows:

- Supplementary information is extended by figures showing

  - profiles of temperature and salinity at $t = 0$;

  - $T_B$ before $t = 0$;

  - results of the ACCESS ESM-1.5 SSP5-8.5 simulation for seafloor temperature in support of our choice for future scenarios of climate change;

  - permafrost layer and MHSZ simulation from 400 kyr B.P. to 0 kyr B.P.

- We dropped out the assumption that MHSZ is an impermeable layer for $CH_4$ transport. As it was expected, this resulted in a larger methane flux at the sediment-ocean interface during the gradual MHSZ degradation and eliminated the pulse release of methane at the end of this process. However, we still discuss the respective results from the previous simulations (because it is a potentially interesting sensitivity study) The former Fig. 5 is moved into the Supplement.

- Some figures are redrawn and restructured. In particular, the former Figs. 2 and 4 and combined into a single Figure (now referred to as Fig. 2). This is done to make it easier to compare the time of disappearance for permafrost and for MHSZ. Other figures are renumbered accordingly.

- Upon revising our paper, we found an error in our calculations: flux $f_{CH_4}$ *was not* multiplied by $K_S$ except that belonging to the pan-Arctic estimates. Now, this error is corrected.

- We found a logical inconsistency in our notations. In particular, the present-day shelf depth (thickness of water layer above the sea floor) was denoted as $H_{\mathrm{D}}$, while temperature at the sediment water interface was referred to as $T_{\mathrm{B}}$. Now, $H_{\mathrm{D}}$ is replaced by $H_{\mathrm{B}}$ to highlight that they point to the characteristics at the same physical surface.

Below, the point-to-point replies to the comments are presented. The original comments are typed in italics, and the replies are typed in regular font.

**General comments**

- *In my opinion, the chief weakness of the paper lies in some of the assumptions made in modelling future permafrost development, specifically the direct application of projected air temperature anomalies as changes to bottom water temperatures. This is akin to removing the agency of the shelf sea water column, including ice, as a mitigator of temperature changes. As a result, the projections made in this paper (TR1000 & TR3000) are like some kind of mixture of subsea and terrestrial: initiated as subsea permafrost and GHSZ, and then forced by air temperature changes applied to the seabed. If I have perhaps misunderstood the method, then its explanation at least needs clarification. I suggest that almost any additional assumption, for example of a temperature offset (akin to a marine n-factor), would be better than directly applying air temperature changes directly to the seabed. This short-cut in the model will make both scenarios warmer than they otherwise ought to be, but to an unknown degree, and leave me suspecting that the base case (TR0, without anomalous warming) may be closest to what we should expect in reality, and not the two other model runs.*

  *Most studies assume that past and current bottom water temperatures are negative for most of the shelf (and esp. the depths under consideration in this paper). The air temperature anomaly that acts as a somewhat arbitrary threshold is therefore the value that somewhat takes this bottom water temperature above the freezing point of the permafrost, or above the temperature that defines the upper limit of the GHSZ zone.*

  *To address these comments, it is necessary to show the forcing temperatures for the past and projected time period. Future forcing is shown in on its own in the supplementary information. Please allow the reader to evaluate how the applied anomaly compares to historical values.*

  Temperature at the sediment top in our simulations is prescribed to be equal to the water temperature near the seafloor when shelf is flooded (in particular, for the whole future period). In turn, when the shelf is under water this temperature is warmer than the near-surface air temperature. Nonetheless, $T_{\mathrm{B}}$ is negative for all time instants $t \leq 0$. In TR0, $T_{\mathrm{B}} \leq 0^{o}C$ for future 100 kyr as well. For other two scenarios, near-surface temperature anomaly $\Delta T_{\mathrm{fut}}$ is added on top of $T_{\mathrm{B}}(t = 0)$ assuming that the thermal perturbation propagates instantly to the sea floor depth.

This clarification is added to supplementary Sect. S3. In addition, our manuscript is extended by new Figs. S1 which shows $T_B$ for the past time interval for different $H_B$. In addition, new Figs. S1 and S2 (the latter is the former Fig. S1) are redrawn with label '$T_B$' instead of '$T_0$' at $Y$-axis for clarity.

We do not agree that TR0 is assumed to be a base scenario, while TR1000 and, especially, TR3000 are more tentative cases. The peak future warming in scenario TR3000 is $8.4^oC$. Even larger warming near the sea floor at the Arctic shelf is simulated by the ACCESS Earth system model (version 1.5) under SSP5-8.5 scenario in year 2300 (see new supplementary Fig. S3). We note that the equilibrium climate sensitivity (ECS) for this model is $3.9^oC$ which is smaller than it is estimated for a number of the CMIP6 models (Table 7.SM.5 in IPCC AR6 WG1). Thus, even larger response may be expected for Earth System Models with larger ECS (this issue deserves a further study).

- *A second consideration that needs to be addressed is the definition of terms. What is treated as "permafrost" in the analysis of results? Does any amount of ice in the sediment result in its being classified as permafrost? Or does temperature play a role? What is meant by survival? Are permafrost and GHSZ surviving if ANY are present in the sediment column? This is perhaps interesting, but may be misleading, and MUST at any rate be explicit. Please add definitions of these terms and perhaps a discussion of the impact of the definition.*

In our simulations, permafrost is assumed to exist in a sediment layer in a given year if the simulated temperature in this layer is below the freezing temperature for this year. While this criterion does not observe water content of this layer, we note that in our simulations the sediment pore space is assumed to be filled either by liquid water or by ice.

When applied to permafrost, term 'survival' means that permafrost layer, which is formed earlier in a given simulation, continues to exist (probably with different thickness). In a similar way term 'survival' is applied to MHSZ. Survivals of permafrost and MHSZ are studied in separate – each is governed by its own criterion (temperature below the freezing threshold for permafrost and temperature-pressure conditions for MHSZ).

Both explanations are added to Sect. 2 of the manuscript.

- *The authors do a good job of presenting the results of their analyses, either as bar graphs comparing some variable of interest (e.g. timing of extinction) or as depth-time cross-sections. It is however important that the results are presented in a way that permit comparison with the work of others. For example, to put their projections of permafrost and GHSZ into context, it would be necessary to also show past permafrost and GSHZ since 400 kyr BP for at least one scenario (e.g. HD 50, G 60 W/m2). In particular, this would allow comparison with the seminal work of Romanovskii – do their results compare, is there a shift to more or less permafrost, a different timing of GHSZ persistence, etc. and how abrupt are the changes*

*expected in the shift to projected values. This has direct relevance to the extent of the GHSZ in their analyses, and the possibility, as Romanovskii describes, of intra-permafrost hydrates migrating upwards following inter-glacial warm times with upwardly migrating lower permafrost boundaries.*
This information is added as a new supplementary Fig. S6. A brief discussion is added to Sect. 2 of the main text. We note that past changes of permafrost and MHSZ follow the Pleistocene glacial cycles with a delay of the order of $10^1$ kyr – similar to that it was inferred by Romanovskii et al. (2005).

**Specific comments**

- *Mention the "locations" that are modelled in the methods to set the reader up for what follows (i.e. Hb 10, 50, 100).*
  This clarification is added to Sect. 2 of the manuscript.

- *What were initial salinity values after spin-up? Are they reasonable?*
  The profiles of temperature and salinity at $t = 0$ are shown in supplementary Fig. S5, and their description is added to the supplementary Sect. S4. In the shallow and intermediate parts of the shelf, salinity drops within few tens of meters. In the shallow shelf, it is about 20 psu at the depth 10 m below the sea floor, and below the depth of 30 m relative to the sea floor $S$ amounts to few per mil. In the intermediate shelf, salinity value 10 psu is reached at the corresponding depth 50 m relative to the sea floor. In the outer shelf, $S$ is markedly larger and, as a whole, is above 15 psupre until the depth 100 m below the sea floor.

- *Abstract, Line 11: replace "Time instants" with "The timing"*
  The wording is revised.

- *Line 16: I am not sure what is meant by the word "instrumental" in describing the effect of warming on MHSZ loss. Do you simply mean "important" or something more specific?*
  This and other instances of "instrumental" are replaced by "important".

- *Introduction, Line 18: you refer immediately to methane hydrates, rather than gas hydrates. I suppose that you assume methane-only in order to use existing stability relationships? Perhaps add a short discussion of how your results might be affected by a mixing of gases in hydrates?*
  A mixture of methane hydrates and hydrates of other species would change the temperature and pressure conditions for the hydrates formation and existence with impacts on time dynamics of such hydrate stability zone. Unaware of the respective conditions, we just mention such a possibility without attempting to quantify it. The respective discussion is added to the last section of the manuscript.

- *Line 23: "so called" is a somewhat pejorative word in English, and does not work in the way that many Russian authors use it. I suggest deleting.*
  The word is deleted.

- *Line 24 & 26: the term "survived" implies something alive and is a dramatic word. I am fine with this, however, it is not clear what you mean by the "survival of PAMH". Do you mean that any hydrates still exist? Or do you mean that a hydrate stability zone still exists? This question comes up throughout the paper for hydrates and for permafrost. What do you mean by the "survival/extinction of permafrost"? No cryotic sediment? No ice?*
  We agree that term "survive" is too dramatic. The first sentence is reformulated as "Both the subsea permafrost and the permafrost- associated methane hydrates (PAMH) are known to exist at the present day, possibly owing to their long, of the order of $10^1$ kyr (Romanovskii et al., 2005; Malakhova and Eliseev, 2017, 2020a), response time scales to temperature anomaly at the top of the sediments." In the second sentence, "survive" is replaced by "not disappear".

- *Line 27: "are projected"*
  The sentence is corrected.

- *Line 29: Yang et al (2014) also wrote about riverine heat flux for North American Arctic rivers (Polar Science 8 (2014) 232-241).*
  This reference is added to the manuscript.

- *Line 40: "aftermath" not used correctly*
  This word is replaced by 'following'.

- *Line 42: I do not know what "This inception" refers to here.*
  "This inception" is replaced by "Next glacial inception".

- *Line 47: "isotope" is sufficient, not "isotopology"*
  The word is changed accordingly.

- *Line 54: replace "These fluxes might become much stronger near the timing of complete local extinction of the permafrost and hydrate layers," with "These fluxes might become much stronger when permafrost and hydrate layers are completely extinguished"*
  The sentence is revised accordingly.

- *Line 56: you say "below" the frozen sediment layer, but as Romanovskii et al show, intrapermafrost hydrates and gas can be expected to develop over glacial cycles, when permafrost thins through thaw from below. "within and below"*
  "And below" is added to the sentence.

- *Line 57: "dissolved" is usually reserved for the incorporation of solids into a solvent. I understand that "dissolution" implies "dissolved", but*

*this word will be confusing for most readers. What happens to hydrates when they destabilize? I do not like "degrade" (which means to lower in elevation) or "decompose" (which implies organic decay). "Destabilize" does not necessarily mean that the hydrates have disappeared, they may still exist in a metastable state. "Decay" is not a bad choice, although similar to "decompose". After considering all alternatives, I feel more accepting of "dissolve". But it will be confusing.*

We agree that the term is somewhat confusing. However, lacking a better choice (which is indicated by the reviewer as well), we are akin to keep this term.

- *Line 58: improper use of "aftermath"*
  The sentence is put in form "with a corresponding pulse release of methane".

- *Line 58-59: I suggest re-formulating this sentence ("Despite the latter phenomenon...") to: "The catastrophic release may be attenuated by the transient existence of pathways through taliks that form below paleo-river channels, lakes and lagoons, especially...".*
  The sentence is changed accordingly.

- *Model and simulations, Line 78: "For the heat diffusion equation, ..." and "is imposed"*
  The sentence is corrected.

- *Line 83: I hope that the magnitude of the possible effect of latent heat is discussed later in the paper!*
  On one hand, the latent heat of hydrate dissociation leads to retardation of their response to climate change. Thus, it would only prolong the MHSZ existence in our simulations. However, on the other hand, accounting for this latent heat would suppress the formation of methane hydrates during Pleistocene glaciations as well. Provided that other things being equal, thinner MHSZ apparently would disappear earlier. The result of these two mutually compensating effects is unclear. The respective note is added to Sect. 4.3.

- *Line 91: "marine" instead of "oceanic"*
  "Oceanic" is replaced by "marine".

- *Line 96 please add "water" to "depth", otherwise it is not clear*
  The word 'water' is added to the sentence for clarification.

- *Please state explicitly that you run 3 locations with varying HD and call them "shallow", "middle shelf" and "outer shelf" — this would make all explanations and figures much more accessible and intuitive*
  This definition is explicitly listed in Sect. 2 of the revised manuscript.

- *Line 101: Is there a difference between $\Delta T_{\text{fut}}$ and $T_{\text{fut}}$? If not, please use the same variable name*

This is the same variable, which is referred to as $\Delta T_{\text{fut}}$. The erroneously omitted $\Delta$ is inserted.

- *Line 101: Since you are combining water temperatures with air temperature anomalies, it is important that you show these data series for the entire modelling period — are they reasonable?*
  These series are reproduced in new Figs. S1 and S2. There is a time interval (from 4 ky B.P. to $t = 0$) for which these two figures overlap. It is technically inconvenient to combine thes two figures into a single one because of the predominant negative temperature anomalies in the past and the predominant positive temperature anomalies in the future.

- *Line 109-114: I understand that these are bracketing or a window of possibilities, but you should make the case that TR1000 and TR3000 are indicative of something possible, i.e. the TR0 is not the most likely scenario (see general comment above)*
  We do not agree that TR3000 and TR1000 are just tentative. Similar near-seafloor warming at the Arctic shelf is simulated by an Earth system model (new Fig. S3). A respective note is added to Sect. 2 of the revised paper.

- *Line 128-135: This paragraph belongs in the Discussion, not in the Methods*
  Both paragraphs are moved to Discussion.

- *Line 136: I prefer "following" rather than "via"*
  The sentence is ameliorated.

- *Results, Permafrost, Line 145: what is "thick" permafrost? Quantify*
  At the shallow and middle this thickness is from 300 m to $1,200$ m. At the outer shelf, it is $\leq 150$ m. The respective note is added to Sect. 3.1.

- *Line 146: When I read the second sentence, I did not under what cases had been defined -- see comment above about more explicitly defining the HD values run and giving them names; again, I still do not know what you mean by "permafrost disappears" — please define explicitly in the methods*
  All definitions are added to the revised manuscript.

- *Line 148: what is "shelf depth"? Water depth or depth in sediment? Relative to what?*
  We mean the present-day thickness of water layer above the sea floor. The definition is added to Sect. 2.

- *Line 152: permafrost does not "melt", it "thaws"*
  This awkward term is corrected (here and at the next instant).

- *Line 153: "independent of", not "from"*
  The sentence is corrected.

- *Line 156: replace "During the most part of..." with "For most of..."*
  The sentence is ameliorated.

- *Line 166: again, I stumble over "shelf depth", but I realize that what is meant is "water depth", correct?*
  Yes, it is. We are sorry for unclear terminology. Now, an explicit definition of this term is in the manuscript (Sect. 2).

- *Line 167: surely the water temperature is very important? How does it figure in? Is it directly a result of water depth?*
  Sure, water temperature is important as well. However, this variable is not varied explicitly in our paper. Rather, it is a function of $H_{\mathrm{B}}$. Thus, the dependence of our results on contemporary shelf depth implicitly includes the respective dependence on initial near-floor water temperature. The corresponding statement is added to Sect. 3.1.

- *Line 171: I do not feel that Archer (2015) obtained "similar" time scales. Please provide the numbers that you find similar, or perhaps choose different wording?*
  We regret to state that Archer's (2015) paper was misunderstood by us, and the statement on similar time scales for permafrost extinction between our manuscript and Archer's paper was erroneous. Now, this statement is excluded from the manuscript. We are thankful for the reviewer for pointing this out.

- *3.2 Methane hydrates stability zone, I feel that the paper would be strengthened by showing the relationship of permafrost and MHSZ distribution relative to each other, at least for the main scenario, which I think is 50/60 – the concept of intra-permafrost gas hydrates should be discussed in this context*
  We combined previous Fis. 2 and 4 into a single figure. Now, it is clearly visible that, for a given $H_{\mathrm{B}}$, $G$, and emission scenario, MHSZ disappears earlier than the respective permafrost layer. However, we did not find a systematic dependence for the difference between these two extinction times on the above-listed parameters. A respective paragraph is introduced at the end of Sect. 3.2.
  Nonetheless, this issue deserves a further study – we are planning to do this in future. In principle, we could estimate the MHSZ thickness $D_{\mathrm{MHSZ}}$ as a function of the permafrost layer thickness $D_{\mathrm{pf}}$ (or vice verse). In our preliminary calculations, future changes (corresponding only to the shrinking of both layers) of these variables follow such functional relationship almost perfectly. However, for past changes the picture is different: they follow a histeresis-like loop. The reason for this is due to long time scales of the subsea permafrost and of the associated MHSZ as reported by Romanovskii et al. (2005), Mestdagh et al. (2017), and by Malakhova and Eliseev (2017). Thus, a brief discussion on this matter would be misleading, and a lengthy one would make our manuscript cumbersome.

- *3.3 Methane release from the sediment to the water, Line 215-225: this really belongs in the methods section; this is where I looked for it when reading the paper: how did the authors calculate fluxes?*
  This paragraph is moved to Sect. 2.

- *Why is "f" sometimes used, and sometimes "F"? is there a difference? If so, define in the methods.*
  We use $f$ for fluxes per unit area (mass per unit area per unit time) and $F$ for the area-summed fluxes (mass per unit time). We agree that these letters were used in a somewhat confusing way in our previous manuscript version. Now this is ameliorated. In addition, a note is added on the difference between $f$ and $F$ as well as on the difference between $m$ and $M$.

- *Line 218: in fact, the saturation limit depends on the rate of delivery of methane to the sulphate reduction zone. Is this consistent with the use of a simple coefficient?*
  Sure, it is. Our usage of a single coefficient to represent sulfate reduction is a drastic simplification. However, because we do not account for explicit geography in our set up, we feel that our estimates are correct at least for the order of magnitude in this respect – it is clear that an apparent, mechanistically-derived $K_{\mathrm{S}}$ can not be larger than unity, and it is likely to be of the same order of magnitude as 1/2. A respective note is added to Sect. 2 of the paper.

- *Line 224-5: should be "...is adapted from Ruppel and Kessler (2017), who synthesized..." and then say what they synthesized.*
  The wording was awkward. Now it is ameliorated. In particular, 'synthesised' is replaced by 'reviewed'.

- *Line 243: It is important that you compare your fCH4 results to available observational data. However, I cannot see any values on Figure 4 that correspond to the values that you report for Shakhova. Please state more explicitly which values of yours are comparable to the range that you quote.*
  The referenced range (up to 10 $\mathrm{TgC_4\,yr^{-1}}$ taking into account impacts of an initial degree of subsea permafrost thaw and by modern methanogenesis combined with partial release of preformed $CH_4$ from inter-pore and/or relic hydrates preserved within the permafrost at the shallow and intermediate shelf) is added to this Figure.
  Please note that previously we used another, much smaller value of the empirically-based estimated for such fluxes (up to 1 $\mathrm{TgC_4\,yr^{-1}}$). The reason for this that we misunderstood the Shakhova et al. (2019) review – we overlooked word 'initial' in their sentence 'permafrost thaw'. Now, this error is corrected, and the empirically-based value to compare with is an order-of-magnitude larger (up to 10 $\mathrm{TgC_4\,yr^{-1}}$). We are sorry for this error.

- *3.4 Implications for the pan-Arctic, Line 247: "rudimentary", not "rudimental"*
  The word is corrected.

- *Line 251: "We assume limit. . . " should be replaced with "We limit. . . "*
  The phrase is revised accordingly.

- *Line 253-4: When you refer here to subsea distribution, do you mean depth, lateral area or both?*
  Upon revision, it is clearly stated that the geographycal distribution is meant.

- *Line 260-1: I am confused by the sentence "This anomaly is apparently different even from temperature in other model grid cells." – it seems expected that the anomaly in the East Siberian Arctic shelf would be different than in other cells?*
  Yes, sure. This paragraph is included into the manuscript to show that we checked *how different* is the anomaly in this grid cell from its counterparts at the same latitudes in Climber-2. We suggest that no revision is needed.

- *Line 264-6: I am not sure what this $-12^oC$ reference temperature is or how it is used. For what is it a reference? I understood from the methods section that the anomaly was added to the water temperatures?*
  This value is only used when the shelf is exposed to the atmosphere. When the shelf is under water, another value ($T_w$) is used. The sentence is clarified.

- *Line 280: replace "could" with "to"*
  The sentence is revised accordingly.

- *Line 284: replace "devote" with "require"*
  This sentence ios removed from the paper upon revision.

- *Line 337: "permafrost disappears", not "permafrost is disappears"*
  The misprint is corrected.

- *Line 338: "a few centuries"*
  The sentence is revised accordingly.

- *Line 353: "rudimentary"*
  All instances of "rudimental" are corrected

- *Line 357: "by up to 2%"*
  The sentence is revised accordingly.

- *Line 361: "depends more weakly on the applied emission"*
  The sentence is revised accordingly.

- *Line 423: "lose" not "loose"*
  The misprint is corrected.

- *Figure 1.*
  *- this would be more effective with the same X/Y axis limits - it is difficult to evaluate these figures without having seen the "big picture": Please include -400 ka to 100 ka for at least 1 scenario, for example HD 50 / G 60*
  Now these figures are redrawn with the same axes limits. The whole scenario is added as supplementary Fig. S6.

- *Figure 3, You model to a depth of 1500 m — it is misleading to have y-axes that extend beyond this depth, and it appears that the base of permafrost was exactly 1500 m. This figure would also work better if all y-axes were the same. At the moment, it makes the impression of equally thick permafrost under all scenarios.*
  The figure is redrawn according the the reviewer's comment.